# Audio Super-Resolution with Latent Bridge Models

**Chang Li**[1,2,3,*] **Zehua Chen**[1,2,*,†] **Liyuan Wang**[1] **Jun Zhu**[1,2,†]

[1]Department of CST, Tsinghua University, Beijing, China
[2]Shengshu AI, Beijing, China
[3]USTC, Hefei, China

```
{zhc23thuml, liyuanwang, dcszj}@tsinghua.edu.cn
              lc_lca@mail.ustc.edu.cn
```

## Abstract

Audio super-resolution (SR), *i.e.*, upsampling the low-resolution (LR) waveform to the high-resolution (HR) version, has recently been explored with diffusion and bridge models, while previous methods often suffer from sub-optimal upsampling quality due to their uninformative generation prior. Towards high-quality audio super-resolution, we present a new system with latent bridge models (LBMs), where we compress the audio waveform into a continuous latent space and design an LBM to enable a *latent-to-latent* generation process that naturally matches the *LR-to-HR* upsampling process, thereby fully exploiting the instructive prior information contained in the LR waveform. To further enhance the training results despite the limited availability of HR samples, we introduce frequency-aware LBMs, where the prior and target frequency are taken as model input, enabling LBMs to explicitly learn an *any-to-any* upsampling process at the training stage. Furthermore, we design cascaded LBMs and present two prior augmentation strategies, where we make the first attempt to unlock the audio upsampling beyond 48 kHz and empower a seamless cascaded SR process, providing higher flexibility for audio post-production. Comprehensive experimental results evaluated on the VCTK, ESC-50, Song-Describer benchmark datasets and two internal testsets demonstrate that we achieve state-of-the-art objective and perceptual quality for *any-to-48*kHz SR across speech, audio, and music signals, as well as setting the first record for *any-to-192*kHz audio SR. Demo at https://AudioLBM.github.io/.

## 1 Introduction

Audio super-resolution (SR) systems aim to generate the high-resolution (HR) audio waveform from the observed low-resolution (LR) waveform [47, 56, 109]. Audio SR plays a crucial roles in various applications, including historical recording restoration [73, 59, 74], hearing aids [30], and improvement of perceptual quality [62]. Previous works have explored audio SR with diverse methods such as mapping-based [60, 59], generative adversarial network (GAN)-based [69, 65], diffusion-based [72, 62], and recently proposed bridge-based [54, 45] methods. However, these efforts focus on the audio SR task in a limited scope, *e.g.*, the speech signals in a benchmark dataset, without generalizing to a broader domain of different audio types. To extend audio SR to multiple domains spanning speech signals, sound effects, and music samples, Liu et al. [62] has recently explored diffusion models in the latent space of mel-spectrogram, enabling *any-to-48 kHz* audio SR. A$^2$SB [45] has recently explored bridge models to synthesize the short-time Fourier transformation (STFT) representation, achieving *any-to-44.1 kHz* music SR.

---

[*]Equal Contribution.
[†]Corresponding Authors: Zehua Chen and Jun Zhu.

39th Conference on Neural Information Processing Systems (NeurIPS 2025).

However, the quality of audio SR at scale is still limited. One major cause is the potential mismatch between their generative frameworks and the audio SR process. As shown in Fig. 2, AudioSR [62] synthesizes the missing part of the mel-spectrogram latent representation from an uninformative Gaussian prior, ignoring the fact that the LR waveform contains informative cues about the HR target. A$^2$SB [45] formulates the audio SR analogous to image inpainting [58], where the removed regions are filled with Gaussian priors that remain uninformative about the target signal. Aiming for a high-quality audio SR system, we propose a latent bridge model (LBM) named AudioLBM, learning the *LR-to-HR* waveform SR process with a *latent-to-latent* generative framework. Specifically, we directly compress the audio waveform into a continuous latent representation, where the latent of the LR waveform demonstrates instructive information for the latent of the HR waveform and avoids the removed areas in previous works [41, 45].

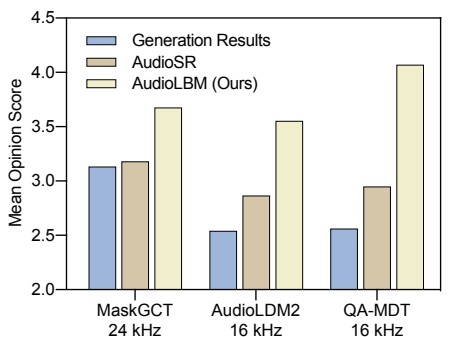

Figure 1: AudioLBM significantly improves the perceptual quality of text-to-speech [111], text-to-audio [63], and text-to-music [53] generation, and outperforms the state-of-the-art *any-to-48 kHz* SR system AudioSR [62].

Then, we design bridge models [14, 130] to enable a *latent-to-latent* generation process, which is inherently matched with the *LR-to-HR* waveform SR task and fully exploits the informative prior in the latent space.

Given the scarcity of high-resolution audio samples, which limits the generalization and scalability of audio SR systems, we propose frequency-aware LBMs that incorporate the prior and target frequencies as explicit conditioning inputs. The awareness of the frequency at both boundary distributions enables AudioLBMs to explicitly learn an *any-to-any* upsampling process across different frequency bands at the training stage, leading to improved upsampling performance to the target resolution at the sampling stage. Furthermore, upsampling the audio signal to a higher sampling rate, such as 96 or 192 kHz, provides enhanced high-frequency detail and allows flexibility in audio post-production. In this regard, we demonstrate that AudioLBM can be naturally extended to unlock the audio upsampling beyond 48 kHz with a cascaded design, where we present the prior augmentation strategies to reduce the cascading error in inference and the fine-tuning techniques to facilitate 192 kHz upsampling.

In summary, we make the following contributions in this work:

- We present a high-quality audio SR system across speech, sound effects, and music samples through modeling the *LR-to-HR* process with a *latent-to-latent* generative framework.
- We propose frequency-aware LBMs and cascaded LBMs, enabling an *any-to-any* training process to overcome data scarcity and empowering SR beyond 48 kHz for the first time.
- Zero-shot *any-to-48 kHz* audio SR evaluated on VCTK [118], Song-Describer-Dataset [68], and ESC-50 [79] demonstrate that our method outperforms prior systems with an average improvement of 21.5% in LSD [25] and 3.05% in ViSQOL [15] across audio and music, as well as achieves state-of-the-art objective and perceptual quality [87] for speech upsampling.

## 2 Related Work

**Audio super-resolution.** Previous works on audio SR can be broadly categorized into spectral-based and waveform-based methods. Spectral-based methods, often referred to as bandwidth extension (BWE), formulate the task as a spectral inpainting problem in either time-frequency space [60, 59, 69, 96, 72, 41, 122, 126, 19, 46] or latent spectral space [62, 37, 110]. In contrast, waveform-based methods directly generate high-resolution signals in the time domain [47, 76, 49, 32, 125, 51, 54]. However, most existing works remain limited in specific domains such as speech or isolated music genres [73, 71], and are evaluated under fixed input resolutions [69, 65, 96], simplified degradation assumptions [32, 55], or limited output sampling rates [72, 46, 69]. These limitations hinder generalization to real-world scenarios involving diverse content and degradation types, such as polyphonic pop music that combines vocals, background instrumentation, and sound effects, or handling extremely low-resolution inputs. Moreover, performing SR beyond 48 kHz (*e.g.*, 96 kHz and 192 kHz) remains unexplored, which provides further benefits in various professional applications,

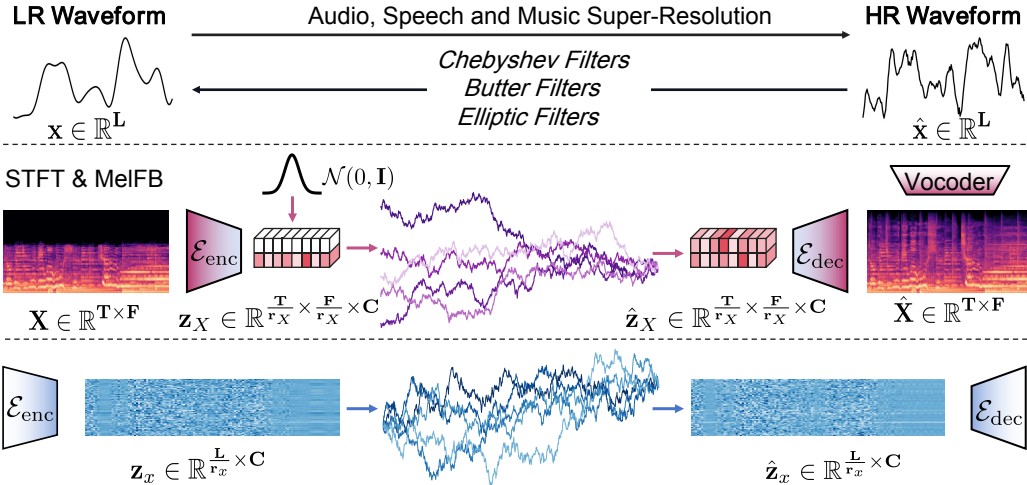

Figure 2: The top part shows how the low-resolution waveform is simulated during training via low-pass filtering, the middle part depicts the baseline method AudioSR [62] that synthesizes high-resolution content from Gaussian noise, and the bottom part presents overview of our proposed AudioLBM. It learns a *latent-to-latent* generation process between the low- and high-resolution waveform latent representations, namely $z_x^{\text{LR}} \in \mathbb{R}^{c_x \times \frac{L}{r_x}}$ and $z_x^{\text{HR}} \in \mathbb{R}^{c_x \times \frac{L}{r_x}}$, where $L$ is waveform length. $c_x$ and $r_x$ are the channel dimension and compression ratio of waveform latent. In contrast, AudioSR operates in latent space $z_X \in \mathbb{R}^{c_X \times \frac{F}{r_X} \times \frac{T}{r_X}}$ of mel-spectrogram $X \in \mathbb{R}^{F \times T}$ with a *noise-to-latent* generation process, where $T$ and $F$ denote time and frequency bins of mel-spectrogram; $c_X$ and $r_X$ denote channel dimension and compression ratio of mel-spectrogram latent.

including mastering [57], post-production [7], spatial audio [8], and immersive content creation [101]. Although AudioSR [62] stands out as a scalable method achieving *any-to-48 kHz* SR across diverse domains, it suffers from two-stage compression pipeline and sub-optimal generative modeling paradigms for the SR task, leading to sub-optimal low-frequency fidelity, misalignment across frequency bands, and high-frequency artifacts [131], which ultimately limit perceptual SR quality. Therefore, building a unified and high-fidelity audio SR model that scales across speech signals, sound effects, and music samples remains a critical challenge. Detailed discussion of audio SR baseline methods is provided in Appendix H.2.2.

**Bridge models.**   Tractable bridge models [130, 14, 64, 52] have received increasing attention for enabling more effective and efficient data-to-data generation paradigms. Unlike diffusion models [34, 98] that generate data by reversing a Markov process that gradually injects noise into the input, bridge models begin from a deterministic and informative prior, and interpolate stochastically toward the target distribution. Recent studies have explored bridge models in various applications, including image translation [52, 103], image restoration [58, 18], dense prediction [38], image editing [11], quality assessment [128, 129] and text-to-speech synthesis [14]. Building on this paradigm, Bridge-SR [54] applies the Schrödinger Bridge (SB) framework directly in the waveform domain for speech SR, using a lightweight WaveNet-based score estimator [106, 44]. In parallel, A²SB [45] employs the SB formulation in the STFT domain for music bandwidth extension and inpainting using a U-Net backbone [89, 22]. However, directly modeling in the data space inherently limits generalization and poses challenges for stable training and flexible scaling to higher-resolution SR. These limitations motivate us to develop a more adaptive generation framework tailored to audio SR.

## 3   Method

In this section, we introduce a latent bridge model (LBM) named AudioLBM to achieve high-quality super-resolution (SR). The key innovations include bridge-based audio upsampling in the continuous latent space of waveform, the frequency-aware model training process, and the cascaded design that unlocks audio SR beyond 48 kHz.

## 3.1 AudioLBM

Considering that the observed LR waveform has been an informative prior of the target HR waveform in the time-domain [54], we train a convolution-based variational autoencoder (VAE) [26] to directly compress the audio waveform into continuous representations, preserving the prior contained in the LR waveform for target generation. The details of VAE architecture and training configurations are introduced in Appendix B. Given the HR audio waveform $x^{\mathrm{HR}} \in \mathbb{R}^L$, we compute the LR counterpart $x^{\mathrm{LR}} \in \mathbb{R}^L$ with various *low-pass filters* to simulate the real-world degradations. Then, we compress them into latent representations $z^{\mathrm{HR}} \in \mathbb{R}^{c \times l}$ and $z^{\mathrm{LR}} \in \mathbb{R}^{c \times l}$ with a pre-trained encoder $\mathcal{E}(x)$, where $l = \frac{L}{r_x}$ and $r_x$ is the down-sampling factor, constructing the boundary distributions and establishing the bridge process as follows.

**Bridge process.** Given $z^{\mathrm{LR}} \in \mathbb{R}^{c \times l}$ as the prior $z_T$ at time step $t = T$ and $z^{\mathrm{HR}} \in \mathbb{R}^{c \times l}$ as the target $z_0$ at time step $t = 0$, we build a bridge process [14] to connect the boundary distributions (see details in Appendix C):

$$z_t = \frac{\alpha_t \bar{\sigma}_t^2}{\sigma_1^2} z_0 + \frac{\bar{\alpha}_t \sigma_t^2}{\sigma_1^2} z_T + \frac{\alpha_t \bar{\sigma}_t \sigma_t}{\sigma_1} \epsilon, \quad \epsilon \sim \mathcal{N}(\mathbf{0}, \mathbf{I}), \tag{1}$$

where $z_t$ denotes the noisy representation at time step $t$ in the forward process of bridge models, and $\alpha_t, \bar{\alpha}_t, \sigma_t, \bar{\sigma}_t$ define the drift and diffusion terms in the associated stochastic differential equation (SDE), thereby controlling the noise schedule of the bridge process. Different from the forward process of diffusion models transforming the clean representation at the beginning $t = 0$ into standard Gaussian noise at the boundary $t = T$, bridge models replace the uninformative Gaussian prior with a Dirac prior $\delta_{z^{\mathrm{LR}}}$, and therefore facilitate the generation process to fully exploit the instructive information contained in the prior distribution *i.e.*, $z^{\mathrm{LR}}$ in AudioLBM.

**Training objective.** There exist multiple equivalent training objectives for bridge models. We empirically find that in the latent space, our AudioLBM achieves higher synthesis quality by using a noise predictor, compared to the data predictor used by recent bridge-related works in waveform [54] or mel-spectrogram domain [14]. In this regard, we formulate the denoising objective as:

$$\mathcal{L}_{\mathrm{bridge}}(\theta) = \mathbb{E}_{z_0 = z^{\mathrm{HR}}, z_T = z^{\mathrm{LR}}, t \sim \mathcal{U}[0, T]} \left[ \left\| \epsilon_\theta(z_t, t, z_T) - \frac{z_t - \alpha_t z_0}{\alpha_t \sigma_t} \right\|_2^2 \right], \tag{2}$$

where $z_t$ is calculated with Eq. (1) at each training iteration.

**Sampling process.** The forward process of bridge models has a reverse process sharing the same marginal distribution $p_t(z_t | z_0, z_T)$ [14, 130]. For signal generation, starting from the prior $z^{\mathrm{LR}}$, we use a first-order SDE-based sampler. From the time step $s$ to the time step $t \in [0, s)$, the first-order discretization gives:

$$z_t = \frac{\alpha_t \sigma_t^2}{\alpha_s \sigma_s^2} z_s + \alpha_t \left( 1 - \frac{\sigma_t^2}{\sigma_s^2} \right) \hat{z}_0(z_s, s) + \alpha_t \sigma_t \sqrt{1 - \frac{\sigma_t^2}{\sigma_s^2}} \epsilon, \quad \epsilon \sim \mathcal{N}(\mathbf{0}, \mathbf{I}). \tag{3}$$

Given a noise predictor $\epsilon_\theta$ well-trained with Eq. (2), we estimate the target $\hat{z}_0$ in Eq. (3) with $\hat{z}_0 = \frac{z_t}{\alpha_t} - \sigma_t \epsilon_\theta$ at each sampling step, and iteratively synthesize the generation target $z^{\mathrm{HR}}$. The pre-trained decoder $\mathcal{D}(z^{\mathrm{HR}})$ of VAE is then used to reconstruct the waveform $x^{\mathrm{HR}}$.

**Bridge process vs. diffusion process.** As shown in Fig. 2, the diffusion-based audio upsampling employs a conditional *noise-to-latent* sampling trajectory, where the prior provides limited information for the target. In comparison, our method employs a *latent-to-latent* sampling trajectory from $z^{\mathrm{LR}}$ to $z^{\mathrm{HR}}$, which has been aligned with the *LR-to-HR* audio upsampling. Moreover, in the continuous latent space directly compressed from the audio waveform, the representation of LR waveform provides instructive information for the generation target, rather than suffering from area removal in a latent space compressed by the STFT representation [45] or mel-spectrogram [122].

## 3.2 Frequency-aware LBMs

Compared to most text-to-speech [111, 24] or text-to-audio [61, 63, 27, 50] generation systems that synthesize the audio samples at a sampling rate below 24 or 44.1 kHz, it is more expensive to collect

the audio samples at a sampling rate of 48 kHz for training audio up-sampling models. To address the limitation of dataset scale, we propose frequency-aware LBMs, enabling an *any-to-any* upsampling process at the training stage. Specifically, given an audio signal $x \in \mathbb{R}^L$, we first filter an HR waveform $x^{\text{HR}} \in \mathbb{R}^L$ with a sampling rate $\text{SR}_{x^{\text{HR}}}$[3] that is lower than the sampling rate of $x$, while it has already provided the frequency band where the audio information predominantly concentrates (detailed in Appendix C). Then, we compute the LR version $x^{\text{LR}} \in \mathbb{R}^L$ with a sampling rate $\text{SR}_{x^{\text{LR}}}$ uniformly sampled from $\mathcal{U}(0, \text{SR}_{x^{\text{HR}}})$, constructing a new LR-HR data pair rather than using a fixed sampling rate for $x^{\text{HR}}$.

At each training iteration, we first compress the $x^{\text{HR}}$ and $x^{\text{LR}}$ into latent $z^{\text{HR}}$ and $z^{\text{LR}}$, and then take their sampling rate $\text{SR}_{x^{\text{HR}}}$ and $\text{SR}_{x^{\text{LR}}}$ as model input, encouraging LBMs to explicitly learn an *any-to-any* upsampling process. Specifically, we extract a sinusoidal embedding of quantized $f_{\text{target}} = \text{Quantize}(\text{SR}_{x^{\text{HR}}}/2)$ and continuous $f_{\text{prior}} = \text{SR}_{x^{\text{LR}}}/2$ and prepend them as two additional tokens of our DiT [78, 4]-based noise prediction network, expanding the feature space to $\mathbb{R}^{c \times (l+2)}$. Towards stronger training performance, we leverage the constant scaling factor $s$ for training bridge models in the waveform domain [54], rescaling the latent with $\tilde{z} = s * z$ for stable training, leading to the training objective of our frequency-aware LBMs:

$$\mathcal{L}_{\text{AudioLBM}}(\theta) = \mathbb{E}_{\tilde{z}_0 = \tilde{z}^{\text{HR}}, \tilde{z}_T = \tilde{z}^{\text{LR}}, t \sim \mathcal{U}[0,T]} \left[ \left\| \epsilon_\theta(\tilde{z}_t, t, \tilde{z}_T, f_{\text{prior}}, f_{\text{target}}) - \frac{\tilde{z}_t - \alpha_t \tilde{z}_0}{\alpha_t \sigma_t} \right\|_2^2 \right], \quad (4)$$

where the noisy latent $\tilde{z}_t$ is computed from rescaled $\tilde{z}_0$ and $\tilde{z}_T$ with Eq. (1). At inference, the detected prior frequency $f_{\text{prior}}$ and the target frequency $f_{\text{target}} = 48 \text{ kHz}/2 = 24 \text{ kHz}$ are both used to condition the model. Specifically, the input waveform is *low-pass filterd* by $f_{\text{prior}}$ to suppress high-frequency artifacts, while $f_{\text{target}}$ guides the model to generate full-band output up to the desired resolution.

### 3.3 Audio SR beyond 48 kHz

**Cascaded LBMs.** Beyond 48 kHz, the audio at an ultra-high sampling rate (*e.g.*, 96 kHz or 192 kHz) provides engineering advantages and post-processing flexibility [85]. However, scaling the upsampling system to such a sampling rate is constrained by the limited capacity of a single model and the scarcity of high-resolution training data, which has not been addressed to date. Here, we propose an extended version of AudioLBM that progressively improves the reconstructed signals from 48 kHz to higher sampling rates. Taking the upsampling to 96 kHz as an example, we introduce cascaded LBMs. We first train a VAE-based compression network $\mathcal{E}(x^{\text{UHR}})$ for the ultra HR audio waveform $x^{\text{UHR}}$, enabling us to design a new AudioLBM $\epsilon_\theta^{\text{UHR}}$ in the latent space $z^{\text{UHR}}$. At generation stage, the waveform $x^{\text{LR}}$ is first upsampled to $x^{\text{HR}}$ with the first-stage LBM, and then taken as the input of the second-stage model, further upsampled to $x^{\text{UHR}}$ with a *latent-to-latent* process

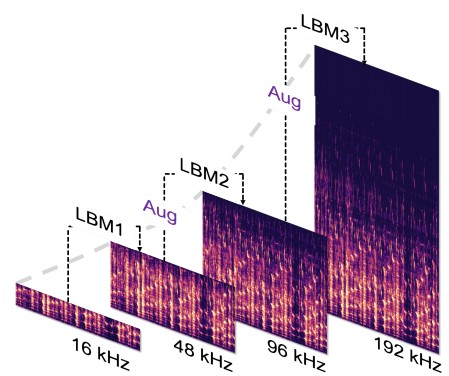

Figure 3: LBMs can be naturally extended into higher-resolution waveform generation with a cascaded paradigm, where prior augmentation is utilized to avoid cascading artifacts and accumulating errors between stages.

from $z^{\text{HR}}$ to $z^{\text{UHR}}$ with Eq. (3), as shown in Figure 3. In the scenario of 192 kHz upsampling, a third-stage LBM is cascaded. However, as the data has been extremely expensive to collect, which restricts the model performance, we propose a fine-tuning method to leverage the pretrained LBM at the second stage, improving the synthesis quality.

**Prior augmentation.** One of the potential limitations of cascaded generative models is the cascading errors in inference [35]. In cascaded LBMs, the LBM at the first stage approximates the ground-truth distribution of $x^{\text{HR}}$, while we inevitably suffer from a mismatch between the synthesized data and the GT signal because of the imperfect learning of $\epsilon_\theta$. Hence, we propose two prior

---

[3]To avoid ambiguity, we define the *low-pass filter (LPF)* by its sampling rate, corresponding to filtering with cutoff at its Nyquist frequency.

augmentation strategies in the waveform and latent space, respectively, for the cascaded bridge models $\epsilon_\theta^{\text{UHR}}$, alleviating the mismatch issue with the first-stage LBM $\epsilon_\theta$ and therefore strengthening the upsampling performance. Firstly, considering the difficulty for generative models to reconstruct fine-grained waveform features, we introduce a degradation operation in the waveform domain. Specifically, in the second upsampling stage, we randomly remove a small portion of high-frequency details near the Nyquist boundary of the prior and obtain a *low-pass-filtered* waveform, denoted as $\text{LPF}(\boldsymbol{x}^{\text{HR}})$. Furthermore, solely simulating the previous stage's output via waveform degradation is insufficient to address potential large-scale artifacts, which may be reflected in unclear harmonics or imbalanced energy distribution. To mitigate this, we further introduce latent-space blurring, which perturbs the audio by applying dynamic Gaussian smoothing to its latent representation along the time axis. Unlike previous diffusion-based [35] or flow-based [95] methods where the latent is corrupted with random noise for augmentation, the blurring strategy provides a deterministic degradation that aligns with the Dirac boundary distributions in bridge models [14], leading to the following training objective for the cascaded bridge model $\epsilon_\theta^{\text{UHR}}$:

$$\mathcal{L}_{\text{CLBM}}(\theta) = \mathbb{E}_{\tilde{\boldsymbol{z}}_0 = \tilde{\boldsymbol{z}}^{\text{UHR}}, \tilde{\boldsymbol{z}}_T = \text{Blur}(\tilde{\boldsymbol{z}}^{\text{HR}}), t \sim \mathcal{U}[0,T]} \left[ \left\| \boldsymbol{\epsilon}_\theta^{\text{UHR}}(\tilde{\boldsymbol{z}}_t, t, \tilde{\boldsymbol{z}}^{\text{HR}}, f_{\text{prior}}, f_{\text{target}}, b_r) - \frac{\tilde{\boldsymbol{z}}_t - \alpha_t \tilde{\boldsymbol{z}}_0}{\alpha_t \sigma_t} \right\|_2^2 \right],$$
(5)

where $\tilde{\boldsymbol{z}}_t$ is calculated with the the ground-truth target $\tilde{\boldsymbol{z}}^{\text{UHR}}$ and the blurred latent prior $\text{Blur}(\tilde{\boldsymbol{z}}^{\text{HR}})$; the condition $\tilde{\boldsymbol{z}}^{\text{HR}}$ is the rescaled latent compressed from the degraded waveform $\text{LPF}(\boldsymbol{x}^{\text{HR}})$; the condition $b_r \sim \mathcal{U}(0, b_r^{\max})$ is the blurring ratio. Hence, the degradation level is also conditioned into the LBM with $f_{\text{prior}}$ and $b_r$. In the training of cascaded LBMs $\epsilon_\theta^{\text{UHR}}$, the small-scale details of the ground-truth prior $\boldsymbol{x}^{\text{HR}}$ has been removed and then the latent prior is blurred, enforcing the model to model the generation of $\boldsymbol{z}^{\text{UHR}}$ from a degraded prior $\text{Blur}(\tilde{\boldsymbol{z}}^{\text{HR}})$ and therefore facilitating the cascaded upsampling process in a robust manner. Further training details of cascaded are provided in Appendix D. At inference time, we select the optimal parameters $b_r^*$ and waveform filtering extent via grid search to maximize cascading quality.

## 4 Experiment

In this section, we first describe the experimental setups and then present the experimental results of *any-to-48 kHz* upsampling and upsampling beyond 48 kHz with an in-depth analysis.

### 4.1 Experimental setup

**Training setup.** We train our compression and SR models on a mixed corpus comprising speech, audio, and music data. All recordings with an original sampling rate below 32 kHz are filtered out, resulting in a total of approximately 5,000 hours of training data. The detailed dataset information is summarized in Appendix G. For each SR stage, all data are resampled to the corresponding target sampling rate and randomly cropped for 5.12-second for the *any-to-48 kHz* and 48→96 kHz models and 2.56-second segments for the 96→192 kHz upsampling stage.

We apply dynamic low-pass filtering to simulate real-world complexity. For the *any-to-48 kHz* stage, the cutoff frequency is sampled from $\mathcal{U}(1{,}000, 20{,}000)$ Hz, the filter type is randomly chosen from {Chebyshev, Butterworth, Bessel, Elliptic}, and the order is sampled from $\mathcal{U}(2, 10)$. For the 48→96 kHz and 96→192 kHz stages, we exclusively use a Chebyshev Type-I filter with order 8, and sample cutoff frequencies from $\mathcal{U}(16{,}000, 48{,}000)$ Hz and $\mathcal{U}(32{,}000, 96{,}000)$, respectively. This design supports arbitrary input sampling rates in any stage and naturally serves as a deterministic waveform-based degradation strategy, as described in Sec. 3.3. For cascading inference, we filter off a frequency range of 4 kHz as waveform degradation and $b_r^{\max} = 1.0$, $b_r^* = 0.3$ for latent degradation for both 96 kHz and 192 kHz SR.

**Evaluation setup.** For the *any-to-48 kHz* task, we randomly sample 500 speech clips from the VCTK [118] test set, 300 audio samples from the ESC-50 fold-5 [79] and 300 music samples from the Song-Describer-Dataset (SDS) [68]. Since both ESC-50 and Song-Describer are natively recorded at 44.1 kHz, we resample all model outputs to 44.1 kHz to ensure a fair comparison. To evaluate performance on native 48 kHz content, we additionally use 300 randomly selected clips from our internal 48 kHz dataset (48Audio). For the 96 kHz and 192 kHz settings, we select 300 audio clips

| VCTK | 8 kHz→48 kHz | | | | | | | | | | |
|---|---|---|---|---|---|---|---|---|---|---|---|
| Metric | Input | UDM+ | NW1* | NW2 | NVSR | Frep | FlowH | APBE | BriSR* | AuSR | Ours | Ours† |
| LSD↓ | 4.069 | 1.232 | 1.417 | 1.141 | 1.003 | 0.907 | 0.816 | 1.003 | 1.047 | 0.940 | 0.753 | **0.742** |
| LSD(L)↓ | 0.187 | 0.216 | 0.268 | 0.294 | 0.357 | 0.272 | 0.194 | 0.224 | **0.172** | 0.486 | 0.773 | 0.708 |
| LSD(H)↓ | 4.456 | 1.345 | 1.544 | 1.239 | 1.085 | 0.985 | 0.889 | 1.093 | 1.143 | 0.994 | 0.724 | **0.712** |
| SSIM↑ | 0.519 | 0.691 | 0.661 | 0.654 | 0.734 | 0.755 | 0.784 | 0.742 | 0.660 | 0.809 | 0.893 | **0.906** |
| SigMOS↑ | 3.136 | 3.068 | 2.936 | 2.838 | 2.831 | 2.743 | 2.792 | 3.082 | 2.998 | 2.846 | 3.023 | **3.095** |

| Metric | Input | AuSR | Ours | Input | AuSR | Ours | Input | AuSR | Ours | Input | AuSR | Ours |
|---|---|---|---|---|---|---|---|---|---|---|---|---|
| **48Audio** | 8 kHz→48 kHz | | | 12 kHz→48 kHz | | | 16 kHz→48 kHz | | | 24 kHz→48 kHz | | |
| LSD↓ | 2.931 | 1.468 | **1.066** | 2.637 | 1.365 | **0.981** | 2.351 | 1.304 | **0.893** | 1.788 | 1.234 | **0.845** |
| ViSQOL↑ | 2.714 | 3.156 | **3.281** | 2.905 | 3.242 | **3.331** | 3.055 | 3.303 | **3.509** | 3.454 | 3.622 | **3.801** |
| **ESC-50** | 8 kHz→44.1 kHz | | | 12 kHz→44.1 kHz | | | 16 kHz→44.1 kHz | | | 24 kHz→44.1 kHz | | |
| LSD↓ | 3.042 | 1.537 | **1.190** | 2.720 | 1.412 | **1.087** | 2.435 | 1.292 | **0.999** | 1.810 | 1.067 | **0.947** |
| ViSQOL↑ | 2.604 | 2.961 | **3.003** | 2.642 | 3.031 | **3.089** | 2.777 | 3.108 | **3.234** | 3.609 | 3.602 | **3.641** |
| **SDS** | 8 kHz→44.1 kHz | | | 12 kHz→44.1 kHz | | | 16 kHz→44.1 kHz | | | 24 kHz→44.1 kHz | | |
| LSD↓ | 4.515 | 1.728 | **1.338** | 4.070 | 1.501 | **1.223** | 3.632 | 1.352 | **1.160** | 1.788 | 1.166 | **1.110** |
| ViSQOL↑ | 1.712 | 2.714 | **2.744** | 1.833 | 2.905 | **2.939** | 2.155 | 3.055 | **3.168** | 3.377 | 3.454 | **3.603** |

Table 1: Objective and perceptual metrics across speech, audio, and music. Baselines marked with *
are our re-implementations. **Ours**† denotes our model trained exclusively on the VCTK training set.

(96/192 Audio) and 300 music (96/192 Music) excerpts from our internal dataset. Each 192 kHz clip
is 2.56 seconds long, while all other evaluation samples are 5.12 seconds.

Long duration inference can be readily implemented using the MultiDiffusion [5, 20, 39] framework,
and we empirically find that employing larger overlap sizes in the early sampling steps, while using
smaller ones but shifting window positions in the later steps (even 0), is more effective in improving
consistency and quality with LBMs.

For objective evaluation, we report the widely used Log-Spectral Distance (LSD) [25] and Spectral
Structural Similarity (SSIM) [113, 59]. To assess perceptual quality, we use ViSQOL [15] for 48 kHz
general audio and music, and SigMOS [87] for 48 kHz speech.

**Baseline method.** For evaluation under the *any-to-48 kHz* setting, unless specifically noted, we
report only our zero-shot results without post-processing[62, 60] for low-frequency replacement. For
speech, in addition to AudioSR (AuSR) [62], we compare against speech-specific baselines trained
on the VCTK-train set, including UDM+ [120], NU-Wave (NW1) [49], NU-Wave2 (NW2) [32],
NVSR [60], FlowHigh (FlowH) [122], Frepainter (Frep) [41], AP-BWE [65], and Bridge-SR
(BriSR*) [54]. We follow the original Bridge-SR configuration, scale up the network to 10.6M
parameters, and retrain it on both the VCTK-train set and our training dataset. For general audio and
music, we compare with AudioSR under the *any-to-48 kHz* configuration. As there are no publicly
available baselines for the 96 kHz and 192 kHz settings, we conduct only ablation studies using our
own models. We scale our *any-to-48 kHz* model to 0.5B parameters, while the two subsequent models
are scaled to 0.3B. All models are trained with an effective batch size of 128 and 1M iterations, while
the fine-tuning procedure described in Sec. 3.3 takes an additional 0.5M steps. Unless otherwise
specified, all ablation experiments are conducted with the same model size, using an effective batch
size of 32 and trained for 0.2M steps without speech data. We use First-Order SDE sampling [14]
for 50 steps for all stages. The compression architecture and additional model details are further
discussed in Appendix B.

### 4.2 *Any-to-48 kHz* upsampling

**Overall performance.** As shown in Table 1, we evaluate our method across speech (48 kHz), gen-
eral audio (48/44.1 kHz), and music datasets (44.1 kHz) using both objective metrics and perceptual
scores. For the 8 kHz→48 kHz speech SR task, probabilistic generative approaches [62, 122] as

| Dataset | ESC-50 (Audio, 16 kHz→48 kHz) | | | | | SDS (Music, 16 kHz→48 kHz) | | | | |
|---|---|---|---|---|---|---|---|---|---|---|
| Metric | w/o Filter | w/o Input-A | w/o Target-A | Ours | only 48 kHz | w/o Filter | w/o Input-A | w/o Target-A | Ours | only 48 kHz |
| LSD↓ | 1.366 | 1.052 | 1.022 | **0.994** | 1.127 | 1.461 | 1.187 | 1.166 | **1.124** | 1.198 |
| LSD-HF↓ | 1.448 | 1.093 | 1.069 | **1.026** | 1.173 | 1.567 | 1.262 | 1.239 | **1.192** | 1.275 |
| SSIM↑ | 0.701 | 0.715 | 0.721 | **0.722** | 0.711 | 0.466 | 0.477 | 0.478 | **0.484** | 0.478 |

Table 2: Ablation studies on ESC-50 and SDS for the 16 kHz→48 kHz SR setting. **Input-A** denotes input-frequency awareness; **Target-A** denotes target-frequency awareness.

| 16→96 kHz on 96Audio and 96Music | | | | 16→192 kHz on 192Audio and 192Music | | | | | |
|---|---|---|---|---|---|---|---|---|---|
| Metric | Input | Direct | Ours | Metric | Input | Direct | 48 kHz | 96 kHz | Ours |
| LSD↓ | 3.068 | 1.406 | **1.216** | LSD↓ | 2.929 | 1.913 | 2.744 | 2.314 | **1.365** |
| LSD (0–48)↓ | 4.006 | 1.498 | **1.083** | LSD (16–96)↓ | 3.191 | 1.452 | 2.304 | 1.372 | **1.160** |
| ViSQOL↑ | 2.562 | 3.010 | **3.330** | LSD (96–192)↓ | 2.880 | 2.244 | 2.879 | 2.879 | **1.474** |

Table 3: Comparison of super-resolution performance on 96 kHz and 192 kHz targets. **Direct** refers to directly trained *any-to-96/192 kHz* models. The **48 kHz** and **96 kHz** columns indicate intermediate outputs in the cascaded pipeline.

well as our method demonstrate clear advantages in capturing global spectral structure (SSIM) and high-frequency quality (LSD(H)). Notably, our zero-shot model significantly outperforms all previous methods in objective evaluation, reducing LSD-HF from FlowHigh's 0.889 to 0.724, improving SSIM from AudioSR's 0.809 to 0.893, and achieving higher perceptual quality (3.023 vs. 2.846 in SigMOS). Due to the domain diversity of the zero-shot training set, we find that the low-frequency noise of speech in some cases is amplified and mistaken for texture in sound effects, which can slightly degrade the perceptual quality. As a point of comparison, we further evaluate a variant trained only on VCTK-train (Ours†), which obtains even better performance and surpasses the GAN-based perceptual SoTA [65] (3.095 vs. 3.082 in SigMOS).

It is worth noting that the methods without compression networks [120, 49, 32, 54] often preserve low-frequency components almost perfectly, as indicated by LSD(L) scores close to the input. However, they often struggle to generate high-quality high-frequency contents. Alternatively, other methods [122, 60, 41, 62] employ post-processing strategies that directly replace the low-frequency band with the original input, thereby achieving near-perfect preservation. In our case, since the model already demonstrates strong overall LSD performance, we refrain from using such replacement strategies (detailed in Appendix E).

**Ablation studies.** We compare our AudioLBMs without filtering any low sampling-rate data (w/o Filter), without any frequency-awareness (w/o Input A), sorely with input-awareness (w/o Target A) and with both input and target frequency awareness (Ours). As shown in Table 2, the use of simple dataset filtering, input-frequency awareness, and output-frequency awareness leads to a steady and progressive improvement in 48 kHz SR performance. This leads to approximately a 20% reduction in LSD and a noticeable increase in SSIM across both audio and music domains, demonstrating the proposed frequency-awareness mechanism under a moderately filtered dataset. Furthermore, the training solely on 48 kHz data significantly reduces the diversity and scale of the training set, leading to sub-optimal performance and a notable drop in both LSD and SSIM. These results further validate the superiority of the proposed *any-to-any* training paradigm.

### 4.3 Upsampling beyond 48 kHz

**Overall performance.** Table 3 presents the comparison results under the 96 kHz and 192 kHz SR settings, where the input waveform is sampled at 16 kHz. We evaluate both cascaded LBMs and directly trained *any-to-96/192 kHz* LBMs. It can be seen that the cascaded LBMs consistently outperform the directly trained counterparts for SR beyond 48kHz. Benefiting from specific trained

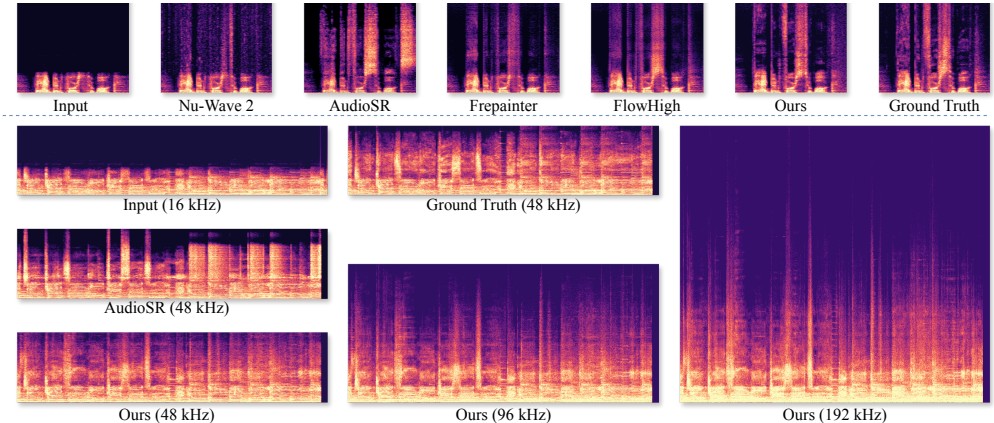

Input    Nu-Wave 2    AudioSR    Frepainter    FlowHigh    Ours    Ground Truth

Input (16 kHz)    Ground Truth (48 kHz)

AudioSR (48 kHz)

Ours (48 kHz)    Ours (96 kHz)    Ours (192 kHz)

Figure 5: For case studies, we present the linear-amplitude STFT spectrograms of a 1.5-second speech segment from the VCTK-test set (sample *p360_102*) and a 5.12-second music clip.

models with sufficient quantity of corresponding audio data, the cascaded 96kHz LBMs achieves better content generation below 48kHz, reflected in a 0.415 reduction in LSD (0–48) and a 0.32 improvement in ViSQOL, which demonstrates the necessity of the cascaded paradigm. Furthermore, a 0.212 reduction in LSD(16-96) under the 192 kHz setting indicates that the proposed prior and conditioning augmentation strategy effectively enhances and refines the content generated by the previous stage.

To validate the effectiveness of the frequency-aware training paradigm in higher-resolution SR scenarios and further investigate the cascading augmentation, we perform extensive ablation studies conducted on the 96Music and 96Audio datasets under the 16→96 kHz SR setting (see Fig. 4). Specifically, we ablate the following components: (i) dataset filtering by removing samples with original sampling rates below 64 kHz, (ii) input/output frequency-awareness, (iii) waveform-domain filtering-based augmentation, (iv) latent-space blurring-based augmentation, and (v) the effect of dynamically sampled blurring ratios during training. Across both datasets, each component consistently contributes to performance improvement, as reflected in the overrall LSD. While the dynamic blur ratio provides additional gains, it also offers the flexibility to support both the cascaded SR pipeline

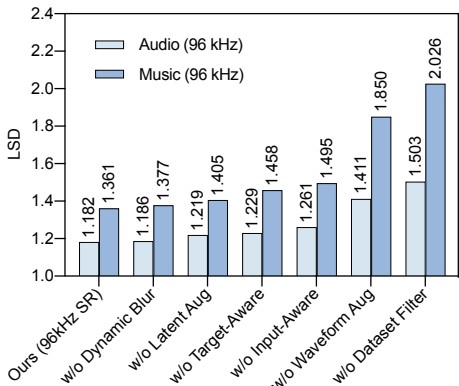

Figure 4: Ablation results on 96Audio and 96Music in the 16→96 kHz setting.

and the standalone 48→96 kHz setting, where no distortion or degradation needs to be applied to the input signal. The SR experiments under real-world scenarios and a detailed comparison of augmentation parameters are provided in the Appendix E.

**Case studies.** Fig. 5 presents case studies on 8→48kHz speech and 16→192kHz music SR using our zero-shot model. For speech, we compare with recent SoTA methods [122, 62, 41, 32]. NU-Wave 2 demonstrates strong low-frequency preservation but struggles to recover high-frequency components under the challenging 6× upsampling setting. Frepainter and FlowHigh capture the general high-frequency contour, yet lack sufficient spectral energy and fine-grained details. AudioSR generates adequate high-frequency content, but often introduces over-boost and undesired artifacts. In contrast, our method achieves detailed full-band reconstruction, maintaining a better energy balance across frequency bands. For music, AudioSR fails to retain low-frequency consistency due to its spectrogram-based design, leading to spectral mismatches and weak harmonic modeling. In contrast, our method takes advantages of LBM modeling and frequency-awareness techniques to generate full-band audio with high fidelity at 48kHz. Further cascading to 96/192kHz introduces clearer high-frequency detail and harmonic structure, confirming the effectiveness of our architecture design and augmentation strategy.

## 5  Conclusion

In this work, we present AudioLBM, a novel audio upsampling system with bridge models in the continuous latent space of audio waveform, modeling the *LR-to-HR* audio upsampling process with a *latent-to-latent* generative framework. By proposing frequency-aware LBMs, we enable the learning of *any-to-any* upsampling process at the training stage to enlarge training data, thereby enhancing the super-resolution quality. We further present cascaded LBMs, empowering the audio upsampling beyond 48 kHz, and propose the prior augmentation strategies to reduce cascading errors. Comprehensive experimental results demonstrate that our AudioLBM outperforms previous audio upsampling systems by a large margin across speech, sound effects, and music signals, and enables high-quality upsampling to 96 kHz and 192 kHz for the first time in audio community. In the future, we plan to extend our super-resolution approach to additional modalities (*e.g.* image, video or time series), and further generalize it to a broader range of restoration tasks. As with other generative audio models, our method may raise concerns about potential misuse, including unauthorized synthesis of speech or music, imitation of artists' vocal identities, and challenges in verifying the authenticity of audio content, which could contribute to misinformation or undermine creative labor.

## 6  Acknowledgements

This work is supported by the National Natural Science Foundation of China (62550004, U24A20342, U25B6003, 92570001).

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

# Appendix

## A Data processing

### A.1 Selecting $f_{\text{target}}$ in *any-to-any* training paradigm

Our *any-to-any* training strategy adopts a flexible target sampling rate $f_{\text{target}}$, where the input rate is randomly sampled as $f_{\text{prior}} \sim \mathcal{U}(f_{\text{min}}, f_{\text{target}})$. While this design improves generalization and robustness, it may also suffer from the presence of unexpected distortion, device artifacts, or environmental noise in high-frequency regions [84], which frequently occur in in-the-wild recordings. As a result, training SR models with $f_{\text{target}}$ exceeding the meaningful frequency range may potentially lead to overfitting to compression artifacts or aliasing patterns, rather than learning useful structure. Such failure cases are shown in Figure 6.

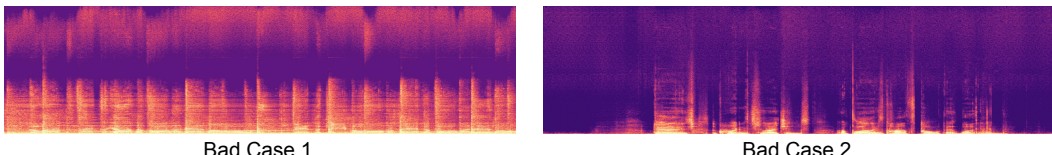

Bad Case 1          Bad Case 2

Figure 6: The impact of high-frequency noise on super-resolution outputs. Without an effective choice of $f_{\text{target}}$, the SR model hallucinates unnatural noise and artifacts in the high-frequency band.

These hallucinations not only degrade perceptual quality [102], but also propagate errors in subsequent cascade stages. To mitigate this, we restrict training targets to a more reliable range by enforcing $f_{\text{target}} \leq f_{\text{eff}}(x)$, where $f_{\text{eff}}(x)$ denotes the estimated effective bandwidth of input $x$ that identifies the frequency range where signal energy is concentrated [10, 124, 102].

**Estimating $f_{\text{eff}}$.** Traditional methods estimate $f_{\text{eff}}(x)$ using energy-based heuristics, such as computing the log-ratio of total signal energy before and after low-pass filtering [112], or applying spectral roll-off thresholds as implemented in Librosa [70] based on cumulative spectral energy. However, these approaches are often sensitive to broadband or high-frequency noise, which can lead to inaccurate estimates of usable bandwidth (see Figure 7). CQT-Diff++ [73] proposes an iterative refinement strategy to more accurately approximate the true effective frequency band. However, it requires carefully tuned filtering parameters and introduces additional inference overhead.

To address these issues, we adopt a robust data preprocessing pipeline, as shown in Figure 8. Given a raw waveform $x$, we first estimate the effective cut-off frequency $f_{\text{eff}}(x)$ using curvature-aware spectral analysis, then apply a low-pass filter at $f_{\text{eff}}(x)$ to suppress high-frequency noise and artifacts, effectively boosting the fidelity and stability of downstream super-resolution models.

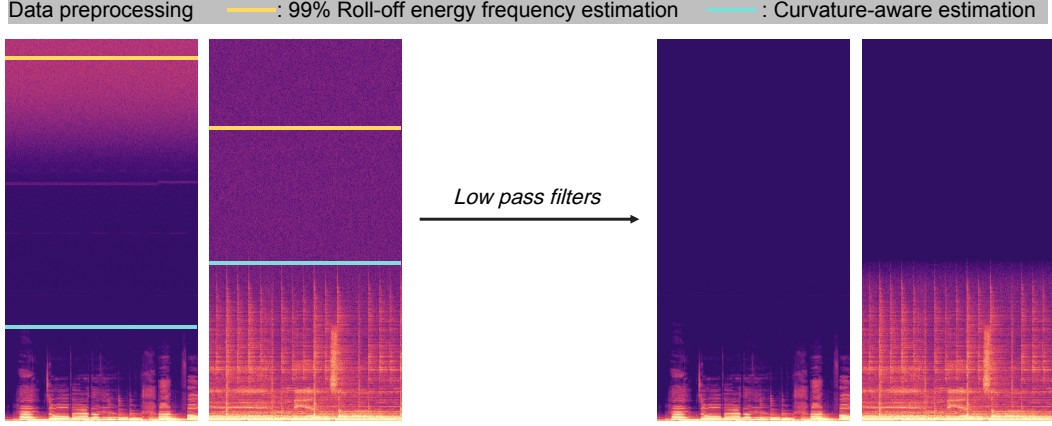

Figure 7: Our proposed data preprocessing pipeline estimates the effective cut-off frequency and removes spectral noise, leading to more suitable samples for SR training.

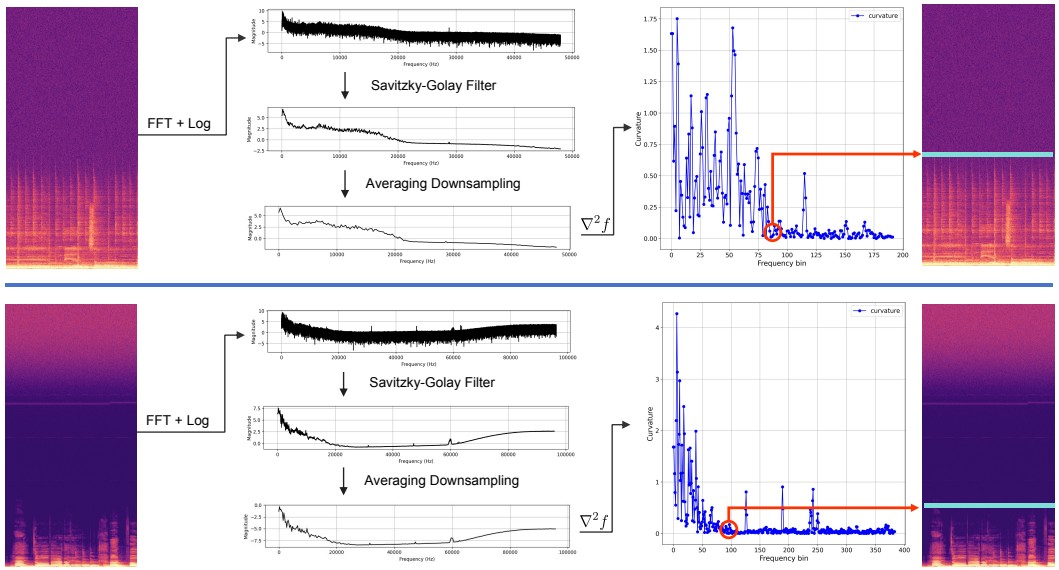

Figure 8: Overview of our proposed curvature-aware estimation pipeline. Given a raw waveform $x$, we first compute its magnitude spectrum via FFT, followed by Savitzky-Golay smoothing and local downsampling to obtain a denoised spectrum. We then compute the second-order curvature over the log-magnitude spectrum to detect the effective cut-off frequency $f_{\text{eff}}(x)$, which is used for downstream low-pass filtering. As a result, our method is robust to high-frequency noise.

We propose a curvature-aware approach to estimate the effective bandwidth $f_{\text{eff}}(x)$ of an audio signal, as illustrated in Figure 8. Our key motivation is that informative frequency regions exhibit stronger local variation, while noise or silence appears smoother—especially in the log-magnitude spectrum.

Given the densely sampled FFT magnitude $X(f) = |\text{FFT}(x)[f]|$, we apply Savitzky-Golay smoothing and downsampling to obtain $\bar{X}_i$, then define the truncation index $i^*$ as the smallest index where local curvature[4] falls below a threshold $\epsilon_{\text{sr}}$, and spectral energy drops below $\tau$:

$$i^* = \arg\min_i \left\{ \max_{j \in [i, i+k]} |\nabla^2 \log \bar{X}_j| < \epsilon_{\text{sr}} \text{ and } \bar{X}_i < \tau \right\}. \tag{6}$$

The estimated bandwidth is then:

$$f_{\text{eff}} = \frac{i^*}{N} \cdot \frac{s_r}{2}. \tag{7}$$

During training, we low-pass filter each input at $f_{\text{eff}}$ to suppress spurious high-frequency components as data preprocessing, improving stability and perceptual quality in subsequent SR stages.

### A.2  Importance of $f_{\text{prior}}$ detection for inference

At inference time, $f_{\text{prior}}(x)$ is estimated for each input sample with the same detection method of $f_{\text{eff}}$ individually, allowing adaptive filtering tailored to the true spectral bandwidth of the audio. This ensures that super-resolution is performed on the informative signal subspace, leading to improved perceptual quality and generalization When generative models are used as the source of input audio, such as MaskGCT [111], the nominal output sampling rate is fixed (e.g., 24 kHz). However, the actual frequency content may vary depending on the speech prompt or training data, and may not occupy the full bandwidth of the input sampling rate. Similar phenomena have also been observed in 16 kHz generation models [61], where the effective bandwidth is often narrower than the nominal sampling rate. Consequently, using the declared sampling rate as the low-pass cutoff for super-resolution input may lead to incorrect reconstruction.

As illustrated in Figure 9, although the model declares a 24 kHz sampling rate, its generated waveform exhibits a clear spectral roll-off well below the Nyquist frequency (12 kHz).

---

[4]$\nabla^2 \log \bar{X}_i \approx \frac{1}{12} \left( -\log \bar{X}_{i+2} + 16 \log \bar{X}_{i+1} - 30 \log \bar{X}_i + 16 \log \bar{X}_{i-1} - \log \bar{X}_{i-2} \right).$

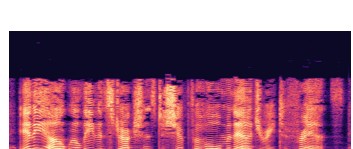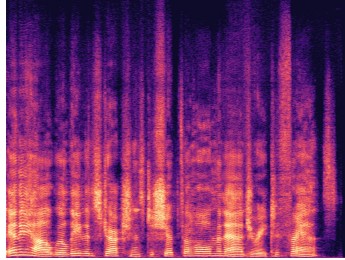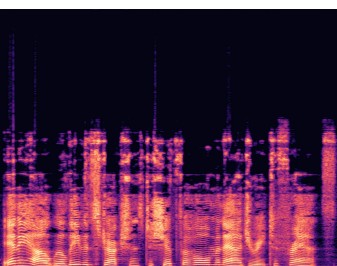

Figure 9: Super-resolution results for samples generated by MaskGCT [111] at 24 kHz, using both AudioSR and our proposed system. While the nominal sampling rate is 24 kHz, the actual spectral content does not span the full bandwidth. As shown in the result of AudioSR, using 24 kHz directly as the cutoff frequency results in incorrect high-frequency generation.

Using 12 kHz as the cutoff frequency by default leads to noticeable spectral discontinuities and artifacts in the output. In contrast, our method first estimates the effective bandwidth of the input audio and adjusts the filtering accordingly, thereby avoiding the generation of spurious high-frequency content and producing a more natural and coherent spectrum.

## B  Compression network

Variational autoencoders (VAEs) [42] include an encoder $\mathcal{E}$ and a decoder $\mathcal{D}$. The encoder maps an input $\boldsymbol{x}$ to a latent representation $\boldsymbol{z} = \mathcal{E}(\boldsymbol{x})$, and the decoder reconstructs $\boldsymbol{x}$ from $\boldsymbol{z}$ via $\hat{\boldsymbol{x}} = \mathcal{D}(\boldsymbol{z})$. The training loss is

$$\mathcal{L}_{\text{VAE}} = \mathbb{E}_{\boldsymbol{x} \sim \mathcal{X}} \left[ \mathcal{R}(\mathcal{D}(\mathcal{E}(\boldsymbol{x})), \boldsymbol{x}) \right] + \text{KL}(q_{\mathcal{E}}(\boldsymbol{z}|\boldsymbol{x}) \,\|\, \mathcal{N}(\boldsymbol{0}, \boldsymbol{I})), \qquad (8)$$

where $\mathcal{R}$ is a reconstruction loss that measures the distance between the original sample $\boldsymbol{x}$ and the reconstructed sample $\mathcal{D}(\mathcal{E}(\boldsymbol{x}))$. $q_{\mathcal{E}}(\boldsymbol{z}|\boldsymbol{x})$ is the approximate posterior distribution over $\boldsymbol{z}$ given $\boldsymbol{x}$, and the KL term regularizes this posterior toward a standard Gaussian prior.

We follow the similar training network structure and pipeline with Stable-Audio-Open [28] and ETTA [50], which combines the following training objectives:

1. A multi-resolution STFT loss [99, 100]:

$$\mathcal{L}_{\text{MRSTFT}}(\boldsymbol{x}, \hat{\boldsymbol{x}}) = \sum_{i=1}^{m} \left( \frac{\|\text{stft}_i(\boldsymbol{x}) - \text{stft}_i(\hat{\boldsymbol{x}})\|_F}{\|\text{stft}_i(\boldsymbol{x})\|_F} + \frac{1}{T} \left\| \log \frac{\text{stft}_i(\boldsymbol{x})}{\text{stft}_i(\hat{\boldsymbol{x}})} \right\|_1 \right),$$

2. An adversarial hinge loss and feature matching loss from EnCodec [21]:

$$\mathcal{L}_{\text{adv}}(\hat{\boldsymbol{x}}, \boldsymbol{x}) = \sum_{k=1}^{K} \left[ \max(0, 1 - D_k(\boldsymbol{x})) + \max(0, 1 + D_k(\hat{\boldsymbol{x}})) \right],$$

$$\mathcal{L}_{\text{feat}}(\boldsymbol{x}, \hat{\boldsymbol{x}}) = \frac{1}{KL} \sum_{k=1}^{K} \sum_{l=1}^{L} \frac{\left\| D_k^l(\boldsymbol{x}) - D_k^l(\hat{\boldsymbol{x}}) \right\|_1}{\text{mean}\left( \left\| D_k^l(\boldsymbol{x}) \right\|_1 \right)},$$

where $D_k^l$ is the $l$-th layer of the $k$-th discriminator $D_k$.

3. The KL divergence loss:
$$\text{KL}(q_{\mathcal{E}}(\boldsymbol{z}|\boldsymbol{x}) \,\|\, \mathcal{N}(\boldsymbol{0}, \boldsymbol{I})).$$

### B.1  48 kHz compression network

An effective compression network is critical to latent-based super-resolution. However, most existing audio compression models fail to maintain strong reconstruction quality, thereby limiting the upper bound of downstream performance. In this section, we analyze the effect of latent compression rate,

KL divergence, and latent scaling coefficient on both reconstruction and generation. Finally, we also compare our VAE to other baselines.

To ensure a fair evaluation of both reconstruction and generation capabilities, we train each ablated VAE configuration on our full speech corpus (details in Appendix G) for 200K steps, and pair it with a corresponding Latent Bridge Model (LBM) trained for 150K steps. All evaluations are conducted on the VCTK-test set. Throughout, we fix the VAE channel dimension to 64 for consistency, as we found that a higher channel dimension often leads to unstable training.

### B.1.1 Compression rate

Typically, for the generation task, higher compression typically improves generative modeling without significantly compromising reconstruction [26]. However, in low-level tasks such as audio super-resolution, we observe that lower compression rates consistently yield better reconstruction and SR performance, as shown in Table 4. Nevertheless, lower compression introduces longer training time and higher inference cost (RTF), thus reducing deployment efficiency.

To this end, we conduct experiments under a fixed LBM training budget (96 A800-hours), and find that a compression rate of 512 achieves the best trade-off between efficiency and performance, delivering consistently superior results across both objective and subjective metrics.

| Metric / Compression rate $r_x$ | 2048 | 1024 | 512 | 256 |
|---|---|---|---|---|
| *Reconstruction* | | | | |
| SSIM↑ | 0.925 | 0.930 | 0.946 | **0.959** |
| LSD↓ | 0.664 | 0.679 | 0.665 | **0.637** |
| SigMOS↑ | 3.099 | 3.083 | **3.169** | 3.142 |
| *Generation* | | | | |
| SSIM↑ | 0.911 | 0.912 | 0.925 | **0.936** |
| LSD↓ | 0.730 | 0.714 | 0.710 | **0.687** |
| SigMOS↑ | 3.088 | 3.025 | **3.139** | 3.088 |
| *Trained in equal time* | | | | |
| SSIM↑ | 0.900 | 0.912 | **0.925** | 0.905 |
| LSD↓ | 0.730 | 0.742 | **0.711** | 0.825 |
| SigMOS↑ | 3.086 | 2.958 | **3.130** | 3.029 |
| RTF (A800, 100 step) | 0.180 | 0.370 | 0.700 | 1.140 |

Table 4: Objective and subjective evaluation under different compression rates $r_x$. Metrics include SSIM, LSD, and SigMOS across reconstruction, generation, and time equalization settings.

### B.1.2 KL divergence

For diffusion-based generation, increasing the KL divergence term is known to improve the alignment between the latent space and the Gaussian prior, thereby enhancing the diffusability of autoencoders and improving generation quality [117, 97]. However, in bridge models, the prior distribution is no longer a standard Gaussian but instead a Dirac distribution induced by the low-resolution waveform latent. In this setting, enforcing KL regularization may no longer be appropriate. Moreover, a smaller—or even zero—KL weight allows the latent space to prioritize reconstruction fidelity, potentially lifting the upper bound of super-resolution quality. To examine this, Table 5 reports results under a fixed compression rate of 512, with KL coefficients set to 0, 1e-7, 1e-5, 1e-3. Consistent with trends observed in Section B.1.1, we find that smaller KL values (as low as 0 or 1e-7) yield better reconstruction and super-resolution performance in the LBM setup—contrasting with the usual observations in diffusion-based generation tasks.

We also observe that reducing the KL weight causes the latent distribution to expand, reflected in significantly increased minimum and maximum values. This can potentially hinder training stability and downstream modeling. To mitigate this, we introduce a simple post-scaling strategy for the latent space. For KL values of {0, 1e-7, 1e-5}, we apply a fixed scaling factor $s$, chosen such

| Metric/KL | 0 | 1e-7 | 1e-5 | 1e-3 |
|---|---|---|---|---|
| *Reconstruction* | | | | |
| SSIM ↑ | **0.942** | 0.942 | 0.940 | 0.935 |
| LSD ↓ | 0.651 | **0.619** | 0.637 | 0.664 |
| pMOS ↑ | **3.111** | 3.079 | 3.103 | 3.030 |
| mean | 0.0506 | 0.1047 | −0.0123 | −0.032 |
| std | 3.9313 | 4.4400 | 1.1670 | 0.9407 |
| min | −27.403 | −29.362 | −8.516 | −4.619 |
| max | 27.615 | 34.446 | 8.506 | 4.774 |
| *Generation* | | | | |
| *s = 0.25* | | | | |
| SSIM ↑ | 0.907 | **0.907** | 0.904 | – |
| LSD ↓ | 0.742 | **0.703** | 0.723 | – |
| pMOS ↑ | **3.095** | 3.060 | 3.073 | – |
| *s = 1.0* | | | | |
| SSIM ↑ | 0.905 | 0.906 | 0.903 | 0.899 |
| LSD ↓ | 0.769 | 0.704 | 0.719 | 0.751 |
| pMOS ↑ | 3.082 | 3.044 | 3.063 | 2.976 |

Table 5: Reconstruction and generation performance under varying KL divergence coefficients (VAE) and latent scaling factors (LBM). Reported on VCTK-test.

that $\text{std}(z \cdot s) \approx 1$, thereby constraining the amplitude range of the latent representation. This normalization consistently improves both objective and subjective metrics.

### B.1.3  Comparison with baselines

Therefore, we set the compression rate to 512, reduce the KL-divergence to 0, and apply a latent scaling factor of $s = 0.25$, resulting in a latent representation **z** with a frame rate of 100 Hz and 64 channels. After training for 1 million steps on the full dataset, we evaluate the reconstruction quality on the ESC-50 and VCTK-test sets and compare it against other baselines. As shown in Table 6, our model achieves the best reconstruction performance both in speech and audio domain across all metrics, which illustrates the advantage of using the compression network with relatively lower compression ratio and KL weight for AudioLBM.

### B.2  Beyond 48 kHz compression networks

In our cascaded architecture, audio super-resolution above 48 kHz is achieved progressively across multiple stages. To avoid making the compression model (VAE) at higher sampling rates a bottleneck, or introducing artifacts that compromise details reconstructed in earlier stages, we use a unified compression ratio of 512 across all sampling rates.

However, high-resolution datasets (e.g., 96 kHz and 192 kHz) are relatively scarce. Training compression models directly on such data can significantly hurt generalization and lead to poorer reconstruction performance. Fortunately, we find that as long as the compression ratio remains consistent, a VAE trained at a lower sampling rate can be directly and effectively reused for higher sampling rates. Specifically, from the perspective of a 96 kHz VAE, 48 kHz audio can be interpreted as a time-compressed (i.e., faster-played) version of high-resolution audio, and thus still represents a valid subset of the high-resolution distribution. Since low-resolution audio data (such as 48 kHz) is more abundant, pretraining compression models on such data provides a strong and robust initialization, even in the presence of some distributional shift. As shown in Table 7, using a compression model pretrained on 48 kHz already yields strong results on the 96Audio dataset. However, despite the much smaller dataset size, directly pretraining at 96 kHz still achieves better performance under the test dataset with native 96 kHz distribution, suggesting a domain gap between the two resolutions. Nevertheless, a lightweight 200k-step fine-tuning on top of the 48 kHz-pretrained VAE surpasses both models, demonstrating strong transferability with only 20% training resource.

| fr | Model | SSIM ↑ | LSD ↓ | LSD-LF ↓ | LSD-HF ↓ |
|---|---|---|---|---|---|
| | | **VCTK (48 kHz)** | | | |
| 100 Hz | agc [3] (continuous) | 0.913 | 0.762 | 0.766 | 0.741 |
| 50 Hz | agc [3] (discrete) | 0.917 | 0.789 | 0.754 | 0.768 |
| 150 Hz | encodec [21] | 0.887 | 1.126 | 0.761 | 1.176 |
| 75 Hz | audiodec [115] | 0.922 | 0.939 | 0.944 | 0.937 |
| 75 Hz | flowdec [114] | 0.925 | 0.619 | 0.560 | 0.630 |
| 100 Hz | **Ours** | **0.951** | **0.580** | **0.428** | **0.602** |
| | | **ESC-50 (44.1 kHz)** | | | |
| 100 Hz | agc [3] (continuous) | 0.730 | 0.836 | 0.748 | 0.840 |
| 50 Hz | agc [3] (discrete) | 0.741 | 0.852 | 0.703 | 0.862 |
| 150 Hz | encodec [21] | 0.704 | 0.925 | 0.910 | 0.922 |
| 100 Hz | dac [48] | 0.734 | 0.799 | 0.735 | 0.801 |
| 75 Hz | audiodec [115] | 0.695 | 0.999 | 0.969 | 0.997 |
| 75 Hz | flowdec [114] | 0.717 | 0.899 | 0.810 | 0.908 |
| 100 Hz | **Ours** | **0.789** | **0.763** | **0.544** | **0.785** |

Table 6: Comparison of reconstruction quality across baselines on VCTK and ESC-50 test sets (resampled to 48 kHz and 44.1 kHz, respectively).

| Step | Phase | Model | fr | SSIM ↑ | LSD ↓ | LSD-LF ↓ | LSD-HF ↓ |
|---|---|---|---|---|---|---|---|
| **96Audio dataset** | | | | | | | |
| 100w | pretrain | 48 kHz-vae | 50Hz | 0.815 | 0.808 | 0.815 | 0.774 |
| 100w | pretrain | 96 kHz-vae | 50Hz | 0.818 | 0.798 | 0.803 | 0.772 |
| 100w | pretrain | 48 kHz-vae | 100Hz | 0.840 | 0.811 | 0.821 | 0.760 |
| 100w | pretrain | 96 kHz-vae | 100Hz | 0.841 | 0.731 | 0.747 | 0.698 |
| 20w | finetune | finetune from 48Hz-vae | 100Hz | **0.850** | **0.709** | **0.703** | **0.692** |
| **192Audio dataset** | | | | | | | |
| 100w | pretrain | 192 kHz-vae | 100Hz | 0.868 | 0.736 | 0.643 | 0.752 |
| 100w | pretrain | 96 kHz-vae | 100Hz | 0.862 | 0.747 | 0.646 | 0.763 |
| 100w | pretrain | 48 kHz-vae | 100Hz | 0.863 | 0.750 | 0.692 | 0.748 |
| 20w | finetune | finetune from 96 kHz-vae | 100Hz | 0.866 | 0.722 | 0.630 | 0.740 |
| 20w | finetune | finetune from 48 kHz-vae | 100Hz | **0.871** | **0.713** | **0.603** | **0.737** |

Table 7: VAE reconstruction results using different training strategies.

We observe similar trends in the 192Audio dataset: models fine-tuned from either 48 kHz or 96 kHz initialization outperform those trained from scratch at 192 kHz. Interestingly, fine-tuning from 48 kHz-pretrained VAE consistently leads to better results compared to fine-tuning from 96 kHz-pretrained VAE, highlighting that the scale of pretraining data matters more than resolution proximity. In other words, pretraining on larger datasets at lower sampling rates yields more transferable and stable compression models for high-resolution audio.

## C  Foundation of bridge models

Bridge models provide a principled and efficient solution for inverse problems and conditional generation in settings where the prior is non-Gaussian yet partially aligned with the target distribution. Given a sample from the prior distribution $x_{\text{prior}}$ and its correspondence from target distribution $x_{\text{target}}$, In score-based generative models (SGMs) [98], a forward SDE is defined between $x_0 = x_{\text{target}} \sim p_{\text{target}}$, and $x_T = x_{\text{prior}} \sim p_{\text{prior}}$:

$$dx_t = f(x_t, t)\, dt + g(t)\, dw_t. \tag{9}$$

Here, $t \in [0, T]$ represents the current time step, with $x_t$ denoting the state of data in the process. The drift is given by the vector field $f$, the diffusion by the scalar function $g$, $w_t$ is the standard Wiener process, and the reference path measure $p^{\mathrm{ref}}$ describes the probability of paths from $p_{\mathrm{prior}}$ and $p_{\mathrm{target}}$.

Under the above framework with boundary constraints, the Schrödinger bridge (SB) problem [94, 14] seeks to find a path measure $p$ of specified boundary distributions that minimizes the Kullback-Leibler divergence $D_{\mathrm{KL}}$ between a path measure and reference path measure $p_{\mathrm{ref}}$:

$$\min_{p \in \mathcal{P}_{[0,T]}} D_{\mathrm{KL}}(p \parallel p^{\mathrm{ref}}) \quad s.t. \quad p_0 = p_{\mathrm{prior}}, \ p_T = p_{\mathrm{data}}, \tag{10}$$

where $\mathcal{P}_{[0,T]}$ denotes the collection of all path measures over the time interval $[0, T]$.

In SB theory [108, 13, 14], this specific SB problem can be expressed as a pair of forward-backwords linear SDEs:

$$dx_t = \left[ f(x_t, t) + g^2(t) \nabla \log \Psi_t(x_t) \right] dt + g(t) dw_t, \tag{11}$$

$$dx_t = \left[ f(x_t, t) - g^2(t) \nabla \log \widehat{\Psi}_t(x_t) \right] dt + g(t) d\overline{w}_t, \tag{12}$$

where the non-linear drifts $\nabla \log \Psi_t(x_t)$ and $\nabla \log \widehat{\Psi}_t(x_t)$ can be described by coupled partial differential equations (PDEs). A closed-form solution for SB [14] exists when Gaussian smoothing is applied with $p_0 = \mathcal{N}(x_{\mathrm{HR}}, \epsilon_0^2 I)$ and $p_T = \mathcal{N}(x_{\mathrm{LR}}, \epsilon_T^2 I)$, to the original Dirac distribution. By defining $\alpha_t = e^{\int_0^t f(\tau)d\tau}$, $\bar{\alpha}_t = e^{\int_1^t f(\tau)d\tau}$, $\sigma_t^2 = \int_0^t \frac{g^2(\tau)}{\alpha_\tau^2} d\tau$, and $\bar{\sigma}_t^2 = \int_t^1 \frac{g^2(\tau)}{\alpha_\tau^2} d\tau$, with boundary conditions and linear Gaussian assumption, tractable form of SB is solved as

$$\widehat{\Psi}_t = \mathcal{N}\left(\alpha_t x, \alpha_t^2 \sigma_t^2 I\right), \quad \Psi_t = \mathcal{N}\left(\bar{\alpha}_t y, \alpha_t^2 \bar{\sigma}_t^2 I\right) \tag{13}$$

under $\epsilon_T = e^{\int_0^T f(\tau)d\tau} \epsilon_0$ and $\epsilon_0 \to 0$ [14]. Therefore, marginal distribution of $x_t$ at state $t$ is also Gaussian:

$$p_t = \Psi_t \widehat{\Psi}_t = \mathcal{N}\left( \frac{\alpha_t \bar{\sigma}_t^2}{\sigma_1^2} x_0 + \frac{\bar{\alpha}_t \sigma_t^2}{\sigma_1^2} x_T, \frac{\alpha_t^2 \bar{\sigma}_t^2 \sigma_t^2}{\sigma_1^2} I \right). \tag{14}$$

# D  Cascaded LBMs

In this section, we provide a detailed description of the cascaded LBM, including analyses of the proposed waveform-based and latent-based augmentation techniques.

## D.1  Waveform-based filtering augmentation

Samples generated from the *any-to-48 kHz* stage often exhibit small-scale high-frequency artifacts, as shown in Figure 10 (a). Directly feeding such artifacts into the next-stage super-resolution model can significantly degrade performance, as shown in Figure 10 (b), where high-frequency content becomes difficult to reconstruct. To address this, we apply Chebyshev Type-I low-pass filters (order=8) with sharp frequency roll-off in the spectral domain, as shown in Figure 10 (c)–(e), removing a frequency content of 8 kHz, 4 kHz, and 2 kHz, respectively, before feeding them into the next-stage model. The resulting outputs are shown in Figure 10 (f)–(h).

As shown in Figures 10 (g) and (h), moderate filtering removes unwanted high-frequency artifacts while preserving useful mid-frequency information, enabling the model to generate high-frequency details more effectively. However, overly aggressive filtering, such as in Figure 10 (g), eliminates too much information. This forces the model to simultaneously reconstruct missing mid-to-low frequency content and synthesize high frequencies, resulting in over-smoothed outputs compared to the result in Figure 10 (h).

The quantitative results presented in Table 8 further validate our findings. We conduct super-resolution experiments on both 48 kHz waveform generated by a preceding 16→48 kHz model (denoted as *Generated*) and 48 kHz audio obtained by downsampling real 96 kHz recordings (denoted as *Real*). For the *Generated* setting, waveform-based augmentation proves consistently effective across all filtering scales, with a moderate filtering ratio achieving the best trade-off between artifact removal and content preservation. Notably, for the *Real* setting, although inference benefits most from models trained without filtering, applying dynamic waveform-based augmentation during training still leads to improved performance. This suggests that waveform-based augmentation remains beneficial in both synthetic and real-data scenarios, highlighting its general applicability and robustness.

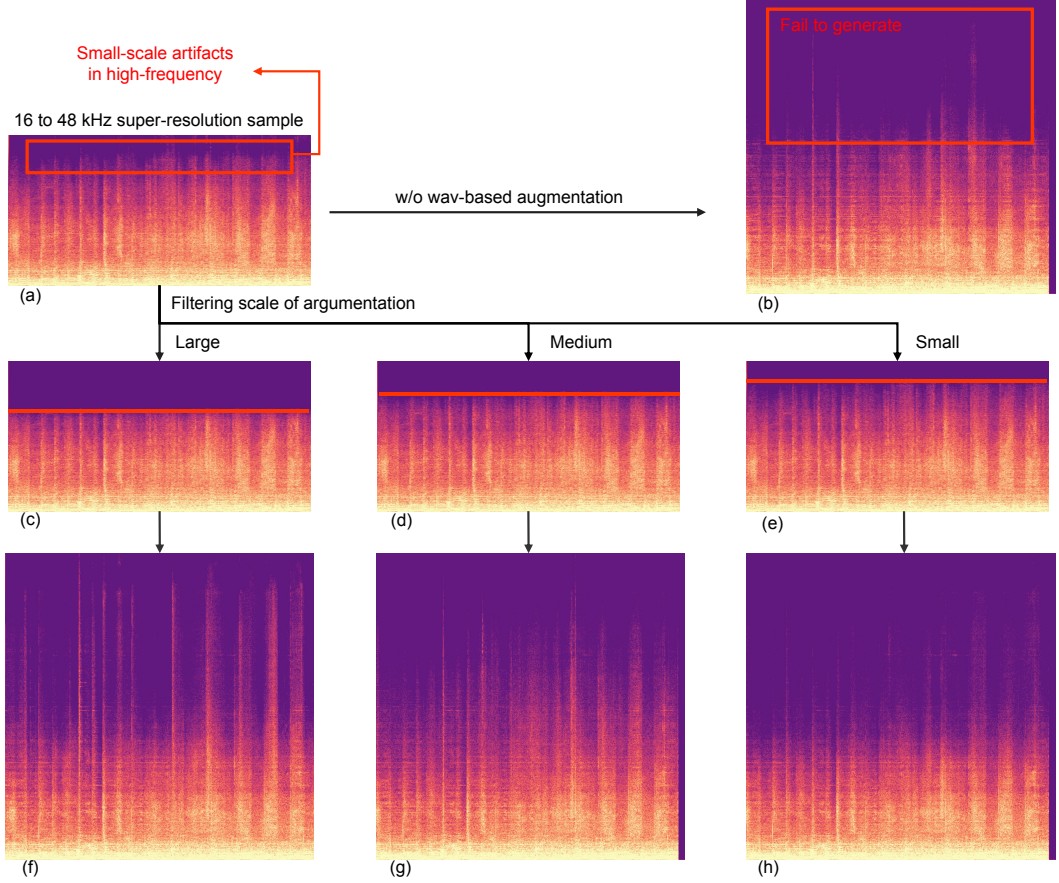

Figure 10: Visual comparison of waveform-based filtering strategies. (a) shows the baseline output from the *any-to-48 kHz* stage with high-frequency artifacts. (b) demonstrates that direct super-resolution from such inputs results in degraded high-frequency reconstruction. (c)–(e) apply *low-pass filters* with different cutoff frequencies, and the corresponding outputs after super-resolution are shown in (f)–(h).

| 48→96 kHz SR | SR on Generated 48 kHz waveform | | | | | SR on Real 48 kHz waveform | | | | |
|---|---|---|---|---|---|---|---|---|---|---|
| Filter Ratio: $24-\delta$ | 18 kHz | 19 kHz | 20 kHz | 22 kHz | w.o. | 12 kHz | 15 kHz | 18 kHz | 21 kHz | w.o. |
| LSD ↓ | 1.532 | **1.495** | 1.520 | 1.534 | 1.518 | 1.147 | 1.193 | 1.122 | **0.978** | 1.282 |
| SSIM ↑ | 0.539 | **0.541** | 0.539 | 0.538 | 0.537 | 0.763 | 0.762 | 0.766 | **0.767** | 0.757 |

Table 8: Ablation study on waveform-based filtering augmentation for 48→96 kHz super-resolution on the 96Music dataset.

## D.2    Latent-based blurring augmentation

To improve the alignment between the predicted and ground-truth latents in the first stage, we introduce a blurring augmentation strategy applied to the latent of the upsampled waveform from previous stage. Specifically, after performing waveform-based degradation—where a small portion of high-frequency spectral content is removed using *low-pass filtering*—we obtain a degraded latent representation $\tilde{z}^{\text{HR}} \in \mathbb{R}^{c_x \times \frac{L}{r_x}}$ .

Specifically, we apply one-dimensional Gaussian blurring to each channel of $\tilde{z}^{\mathrm{HR}}$, which helps smooth local fluctuations and provides a stable initial point for bridge sampling:

$$z_{\mathrm{prior}}[c](t) = \mathrm{Blur}(\tilde{z}^{\mathrm{HR}}) = \sum_{\tau=-k}^{k} w(\tau)\,\tilde{z}^{\mathrm{HR}}[c](t-\tau), \quad w(\tau) = \frac{1}{Z}\exp\left(-\frac{\tau^2}{2b_r^2}\right), \qquad (15)$$

where $c \in [1, c_x]$, $b_r$ denotes the blur ratio, $k$ is half the kernel size, and $Z$ is the normalization constant ensuring $\sum_\tau w(\tau) = 1$. We choose a kernel size of 5, which is significantly larger than $3b_r$, ensuring sufficient coverage of the Gaussian support. During training, the blur ratio $b_r$ is uniformly sampled from the range $(0, 1)$.

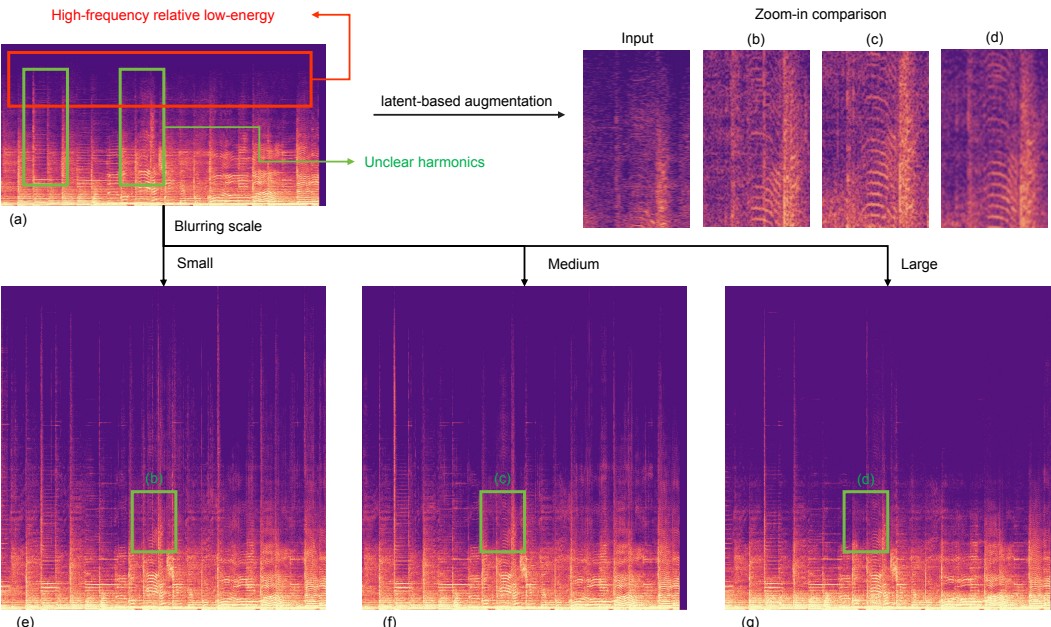

Figure 11: Visual comparison of latent-based blurring strategies. (a) shows the baseline output from the *any-to-48 kHz* stage, where unclear harmonics and low high-frequency energy can be observed. (e)–(g) illustrate the effect of applying different blur ratios $b_r = 0.05, 0.3, 0.5$ to the prior latent during bridge sampling. (b)–(d) show zoom-in views of (e)–(g) respectively for closer inspection.

As shown in Figure 11, in addition to the small-scale high-frequency artifacts described in Section D.1, the generated samples may also exhibit unclear harmonics or relatively low energy in the high-frequency band—both of which can negatively affect downstream stages and lead to cascading errors.

To address this, we apply different degrees of prior blurring with $b_r = 0.05$, $b_r = 0.3$, and $b_r = 0.5$, as shown in Figures 11 (e)–(g), with zoom-in comparisons in (b)–(d). The results demonstrate that latent-based augmentation effectively mitigates the imperfections in the input prior and improves overall reconstruction quality. It not only enhances high-frequency generation but also improves low-frequency reconstruction, leading to more balanced spectral energy and clearer harmonic structures.

Among them, Figure 11 (f) (with $b_r = 0.3$) yields the best result. A smaller blur ratio, as in Figure 11 (e), may not sufficiently correct low-frequency inconsistencies, while a larger blur ratio, as in Figure 11 (g), may overly smooth useful details, resulting in insufficient high-frequency generation. As shown in Table 9, applying latent-based blurring significantly improves performance over the no-augmentation baseline. A moderate blur ratio ($b_r = 0.3$ for generated, $b_r \to 0$ for real inputs) yields the best trade-off, effectively reducing high-frequency artifacts while preserving essential structure. From the results on *SR on Real 48 kHz waveform*, we observe a similar trend as discussed in Section D.1: applying stronger blurring generally leads to worse performance on real data, as low-frequency content is already accurate and does not require correction. This suggests that during inference, only a small or near-zero blur ratio is needed for real 48 kHz data. Nevertheless, applying a

| 48→96 kHz SR | SR on Generated 48 kHz waveform | | | | | SR on Real 48 kHz waveform | | | | |
|---|---|---|---|---|---|---|---|---|---|---|
| Blur Ratio: $b_r$ | 0.4 | 0.3 | 0.25 | 0.20 | w.o. | 0.35 | 0.25 | 0.15 | 0.05 | w.o. |
| LSD $\downarrow$ | 1.399 | **1.361** | 1.377 | 1.380 | 1.405 | 1.196 | 0.984 | 0.978 | **0.976** | 0.995 |
| SSIM $\uparrow$ | 0.531 | 0.542 | **0.543** | 0.538 | 0.540 | 0.735 | 0.769 | 0.771 | **0.771** | 0.759 |

Table 9: Ablation study on latent-based blurring augmentation for 48→96 kHz super-resolution on the 96Music dataset.

mild level of blurring during training still outperforms the no-blur baseline (yielding a 0.02 reduction in LSD and a 0.01 improvement in SSIM), which may be because such operation promotes better understanding of low-frequency content, thereby enabling the model to generate high-frequency components more effectively.

# E  Additional results

In this section, we first continue our discussion on the cascaded stage and present a comparison of several additional augmentation strategies and their effects. We then analyze the role of the low-frequency replacement technique, comparing its use in our system and in other baseline methods.

## E.1  Other augmentation techniques

We additionally compare three alternative augmentation strategies: (1) *latent noise*, where $z_{\text{prior}} = \tilde{z}^{\text{HR}} + n_r \cdot \mathcal{N}(0, \boldsymbol{I})$; (2) *pixel blur*, where $x_{\text{prior}} = \text{Blur}(\tilde{x}^{\text{HR}})$; and (3) *pixel noise*, where $x_{\text{prior}} = \tilde{x}^{\text{HR}} + n_r \cdot \mathcal{N}(0, \boldsymbol{I})$.

| **Method** | latent blur (fixed) | latent blur (dyn) | pixel blur | latent noise | latent noise | latent noise | latent noise | pixel noise | w/o |
|---|---|---|---|---|---|---|---|---|---|
| $b_r/n_r$ | 0.3 | 0.3 | 0.1 | 0.1 | 0.2 | 0.3 | 0.4 | 0.1 | – |
| LSD $\downarrow$ | 1.186 | **1.182** | 1.234 | 1.381 | 1.389 | 1.385 | 1.394 | 3.461 | 1.219 |
| SSIM $\uparrow$ | 0.733 | **0.738** | 0.731 | 0.728 | 0.728 | 0.725 | 0.725 | 0.397 | 0.734 |

Table 10: Ablation study comparing different prior augmentation strategies for bridge-based super-resolution. The second row indicates the blur/noise ratio ($b_r/n_r$) used in each method.

We observe that latent blurring consistently outperforms other strategies, especially when using dynamic blur ratios during training. Pixel-level noise introduces instability and fails to converge, while random latent noise degrades gradually with increased noise levels. This suggests that randomized augmentation may not be suitable for bridge models, while deterministic methods such as blurring offer more stable and reliable guidance. Compared to pixel blurring, latent blurring provides a more global and structured form of regularization, which better aligns the low-frequency components with the ground-truth during training.

## E.2  Low-frequency replacement

Low-frequency replacement is a widely adopted post-processing technique in super-resolution systems, particularly in super-resolution models based on compressed representations [60, 62, 122, 37]. Table 11 presents the results of applying post-processing within our cascaded architecture. Specifically, we apply low-frequency replacement to the output of the any-to-48 kHz model by substituting the sub-16 kHz frequency band with the corresponding band from the input waveform, resulting in the **48 kHz post** variant. This serves as the input for the subsequent 48→96 kHz super-resolution stage, producing **96 kHz post**. Finally, a second round of low-frequency replacement is performed on the output waveform to yield the **Final post** result.

From the table, we observe that performing post-processing at the 48 kHz stage significantly improves low-frequency fidelity, as reflected by improvements in LSD (0–48 Hz: 2.714 → 2.437) and ViSQOL

| Band (Hz) | Metric | Direct | Cascaded (w/o post) | 48 kHz Upsampling | 48 kHz Post | 96 kHz Post | Final Post |
|-----------|--------|--------|---------------------|-------------------|-------------|-------------|------------|
| 0–48 | LSD ↓ | 3.785 | 1.552 | 2.714 | 2.437 | 1.314 | **1.291** |
| 0–16 | LSD-LF ↓ | 0.186 | 0.485 | 0.811 | **0.331** | 0.522 | 0.342 |
| 16–48 | LSD-HF ↓ | 5.071 | 1.738 | 1.614 | 1.596 | 1.221 | **1.220** |
| 48–96 | LSD-SF ↓ | 3.329 | 1.603 | 3.532 | 3.141 | 1.507 | **1.487** |
| 0–48 | ViSQOL ↑ | 1.954 | 2.602 | 3.073 | **3.144** | 3.119 | 3.137 |

Table 11: Ablation study of low-frequency replacement as a post-processing technique for 16→96 kHz super-resolution on the 96Music dataset. **Direct** denotes *any-to-96 kHz* model; **Cascaded (w/o post)** denotes the raw output from cascaded LBMs without replacement; **Post** indicates low-frequency replacement at corresponding stages.

| Method | Modeling Space | Method Type | RTF ↓ | NFE |
|--------|----------------|-------------|-------|-----|
| Ours (any → 48 kHz) | Wav-VAE | Bridge | 0.369 | 50 |
| Ours (any → 96 kHz) | Wav-VAE | Cascaded Bridges | 0.695 | 100 |
| Ours (any → 192 kHz) | Wav-VAE | Cascaded Bridges | 1.351 | 150 |
| Bridge-SR [54] 48 kHz | Waveform | Bridge | 1.670 | 50 |
| UDM+ [60] 48 kHz | Waveform | Unconditional Diffusion | 2.320 | 100 |
| AudioSR [62] 48 kHz | Mel-VAE | Conditional Diffusion | 0.948 | 50 |
| Fre-painter [41] 48 kHz | Mel | GAN | 0.009 | 1 |
| NVSR [60] 48 kHz | Mel | GAN | 0.033 | 1 |

Table 12: Real-time factor (RTF) results on an NVIDIA-A800 under the 48 kHz setting, as well as several baselines.

(3.073 → 3.144). More importantly, starting the 48→96 kHz super-resolution from a more accurate low-frequency foundation leads to better high-frequency generation, shown by the LSD-SF (48–96 Hz) improvement from 1.603 to 1.507.

Although the 48→96 kHz stage may slightly degrade the low-frequency components again, we find that performing an additional round of low-frequency replacement on the final 96 kHz waveform helps recover this degradation. As a result, we achieve a comparable overall fidelity in the 0–48 kHz band (ViSQOL: 3.137 vs. 3.144), confirming the effectiveness of post-processing in both any-to-48 kHz and cascaded LBMs settings.

### E.3 Inference computational cost

To comprehensively evaluate the efficiency, we report the real-time factor (RTF) on an NVIDIA-A800 and several baselines under the 48 kHz setting in Table 12.

It is worth noting that under the 48 kHz setting, although our method is not as fast as GAN-based approaches such as [60, 41], due to the iterative sampling nature, it significantly outperforms other iterative-based methods including [54] and the previous state-of-the-art system AudioSR [62].

When scaling to higher sampling rates (e.g., 192 kHz), it naturally leads to slower inference speeds. We mitigate this impact by adopting a lighter network architecture (see Appendix G.2), so the inference speed does not increase linearly with the sampling rate. As shown in Table 12, even when upsampling to 192 kHz, our system is still faster than the 48 kHz waveform-domain bridge system [54].

### E.4 Modeling space and other probabilistic methods

To isolate the contribution of the latent-bridge formulation, we conducted additional experiments using the same speech datasets (OpenSLR [43], Expresso [75], EARS [86], and VCTK-Train [118]) for training all models, and evaluated them on the VCTK-Test set. We compare the following variants without any additional modules, each trained with 500K steps on the any-to-48 kHz setting and tested on the 8-to-48 kHz SR setting with 50-step sampling:

| Method | Modeling Space | SSIM ↑ | LSD ↓ | LSD-LF ↓ | LSD-HF ↓ | SigMOS [87] ↑ |
|---|---|---|---|---|---|---|
| Diffusion | Mel-VAE latent | 0.809 | 0.940 | 0.486 | 0.994 | 2.846 |
| Rectified Flow | Mel | 0.784 | 0.816 | 0.194 | 0.889 | 2.792 |
| Bridge | Complex STFT | 0.809 | 1.295 | 0.414 | 1.401 | 2.951 |
| Bridge | Raw waveform | 0.660 | 1.037 | 0.184 | 1.101 | 2.896 |
| Rectified Flow | Ours Wav-VAE latent | 0.880 | 0.751 | 0.793 | 0.722 | 2.892 |
| Diffusion | Ours Wav-VAE latent | 0.879 | 0.758 | 0.806 | 0.728 | 2.741 |
| Bridge (Ours) | Ours Wav-VAE latent | **0.907** | **0.742** | **0.708** | **0.712** | **3.095** |

Table 13: Comparison of different generative paradigms under the 8-to-48 kHz SR setting. All models are trained on the same datasets for 500K steps.

- Latent Diffusion (on Wav-VAE latents)
- Latent Rectified Flow (on Wav-VAE latents)
- Bridge-STFT (based on the Nemo [33] framework)
- Bridge-Waveform (based on Bridge-SR [54])
- Latent Bridge (Ours)

As shown in Table 13, our latent-bridge model achieves higher SSIM, lower LSD (objective quality), and better SigMOS (subjective quality), supporting that the latent-domain bridge formulation contributes significantly to the final performance, even when controlling for model architecture and training data.

# F   Inference techniques

In this section, we will discuss the low-pass filtering preprocessing method designed to handle real-world downsampled data.

## F.1   Filtering preprocessing for real-world data

| Metric | Cheby1 | Ellip | Bessel | Butter |
|---|---|---|---|---|
| LSD ↓ | **0.977** | 0.996 | 1.193 | 1.017 |
| LSD-LF ↓ | 0.814 | 0.814 | 0.811 | **0.810** |
| LSD-HF ↓ | **0.989** | 1.011 | 1.236 | 1.036 |
| SSIM ↑ | **0.693** | 0.690 | 0.664 | 0.686 |

Table 14: Comparison of different low-pass filters applied for super-resolution inference. Metrics are reported on 16→48 kHz super-resolution.

Even though we adopt various randomized low-pass filters during training to simulate real-world low-resolution audio, we empirically find that it is beneficial to additionally apply a low-pass filter to the upsampled low-resolution waveform during inference. In particular, the filter should ideally exhibit a sharp roll-off in the frequency domain.

As shown in Table 14, we compare four classical low-pass filters—Chebyshev Type I, Butterworth, Bessel, and Elliptic—for post-upsampling filtering, and evaluate their effects on 16→48 kHz super-resolution performance on the ESC-50 test set.

We observe that both the Chebyshev and Elliptic filters lead to better super-resolution results, as they feature faster attenuation in the stopband. In contrast, Butterworth and Bessel filters exhibit slower roll-off. We hypothesize that this is because the model may struggle to distinguish whether the observed high-frequency attenuation originates from the true characteristics of the data or from the filtering artifacts introduced during preprocessing. In the former case, the model may mistakenly learn to continue the attenuation trend, which can impair its ability to effectively generate high-frequency components.

# G  Detailed experiment settings

## G.1  Training dataset

| Sampling Rate | Dataset | Duration | Type |
|---|---|---|---|
| 48 kHz | VCTK-train [118] | 40h | Speech |
| | OpenSLR [43] | 190h | Speech |
| | EARS [86] | 100h | Speech |
| | Expresso [75] | 20h | Speech |
| | MusDB18 [82] | 10h | Music |
| | Medleydb [6] | 10h | Music |
| | [InternalMusic] | 2000h | Music |
| | FSD50K [29] | 100h | Sound |
| | [InternalSound] | 2000h | Sound |
| 96 kHz | [InternalMusic] | 90h | Music |
| | [InternalSound] | 150h | Sound |
| 192 kHz | [InternalMusic] | 5h | Music |
| | [InternalSound] | 10h | Sound |

Table 15: Overview of training datasets used at different sampling rates, including both public and internal sources.

To support multi-rate audio super-resolution, we curate a diverse set of training datasets spanning speech, music, and sound domains across three sampling rates: 48 kHz, 96 kHz, and 192 kHz. As shown in Table 15, As shown in Table 15. We can also observe that high-resolution data becomes increasingly scarce as the sampling rate increases.

## G.2  Model architecture

We adopt the Diffusion Transformer (DiT) architecture as the noise predictor for our bridge model, following the design of Stable Audio Open [28]. During training, we employ the Bridge-gmax schedule, which has been proven effective in previous super-resolution and text-to-speech tasks [54, 14]. The network architecture and training configurations for the three-stage cascade are detailed below.

*Any-to-48 kHz Stage*

**Model:**

- Depth: 24 layers
- Attention heads: 24
- Hidden dimension: 1152
- Scaling factor: 0.25
- Training sample length: 245760 samples (5.12s)

**Training: Model:**

- Batch size: 128
- Optimizer: Adam with $\beta_1 = 0.9$, $\beta_2 = 0.99$
- Learning rate: $1 \times 10^{-5}$
- Weight decay: 0
- Training sample rate range: 2 kHz to 32 kHz
- Bridge $g_{\max}^2 = 1.0$
- Bridge $g_{\min}^2 = 0.001$

***48-to-96 kHz Stage* and *96-to-192 kHz Stage***

**Model:**

- Depth: 16 layers
- Attention heads: 16
- Hidden dimension: 1152
- Scaling factor: 1.00
- Training sample length: 245760 samples (5.12s for 96 kHz 2.56s for 192 kHz)

**Training:**

- Batch size: 128
- Optimizer: Adam with $\beta_1 = 0.9$, $\beta_2 = 0.99$
- Learning rate: $1 \times 10^{-5}$
- Weight decay: 0
- Training sample rate range for *48-to-96 kHz*: 32 kHz to 96 kHz
- Training sample rate range for *96-to-192 kHz*: 64 kHz to 192 kHz
- Bridge $g_{\max}^2 = 1.0$
- Bridge $g_{\min}^2 = 0.001$

## G.3 Evaluation metrics

**Log-Spectral Distance (LSD)**    Log-Spectral Distance (LSD) [25] is a widely used metric for evaluating the performance of audio super-resolution methods. Given a signal $s$ and its super-resolved counterpart $\hat{s}$, their Short-Time Fourier Transforms (STFT) are computed as $S = \text{STFT}(s)$ and $\hat{S} = \text{STFT}(\hat{s})$, where both $S, \hat{S} \in \mathbb{R}^{F \times T}$, with $F$ denoting the number of frequency bins and $T$ the number of time frames. LSD is defined as:

$$\text{LSD}(S, \hat{S}) = \frac{1}{T} \sum_{t=1}^{T} \sqrt{\frac{1}{F} \sum_{f=1}^{F} \left[ \log_{10} \left( \frac{S(f,t)^2}{\hat{S}(f,t)^2} \right) \right]^2}. \tag{16}$$

Here, $S(f, t)$ and $\hat{S}(f, t)$ denote the spectral magnitude at frequency $f$ and time $t$ for the original and super-resolved signals, respectively. LSD penalizes spectral discrepancies, and lower values indicate better perceptual similarity in the frequency domain. To evaluate performance over a specific frequency band $[f_1, f_2]$, we define a band-limited version of LSD as:

$$\text{LSD}_{[f_1, f_2]}(S, \hat{S}) = \frac{1}{T} \sum_{t=1}^{T} \sqrt{\frac{1}{f_2 - f_1 + 1} \sum_{f=f_1}^{f_2} \left[ \log_{10} \left( \frac{S(f,t)^2}{\hat{S}(f,t)^2} \right) \right]^2}. \tag{17}$$

**Structural Similarity (SSIM)**    While LSD is a point-wise spectral metric, Structural Similarity (SSIM) [113] overcomes its limitations by incorporating structural and contextual information. Originally designed for perceptual image quality assessment, SSIM compares local statistics of luminance, contrast, and structure. In the context of audio, we apply SSIM on log-magnitude spectrograms to capture spectral texture fidelity.

The SSIM score between reference $S$ and super-resolved $\hat{S}$ is computed as:

$$\text{SSIM}(S, \hat{S}) = \sum_{k=1}^{K} \left( \frac{(2\mu_{S_k}\mu_{\hat{S}_k} + \epsilon_1)(2\text{Cov}(S_k, \hat{S}_k) + \epsilon_2)}{(\mu_{S_k}^2 + \mu_{\hat{S}_k}^2 + \epsilon_1)(\sigma_{S_k}^2 + \sigma_{\hat{S}_k}^2 + \epsilon_2)} \right). \tag{18}$$

Here, $S_k$ and $\hat{S}_k$ denote the $k$-th local $7 \times 7$ block in the spectrograms, $K$ is the total number of blocks, and $\epsilon_1 = 0.01$, $\epsilon_2 = 0.02$ are stability constants following VoiceFixer [24]. Higher SSIM values imply better preservation of spectral structure.

**Virtual Quality Objective Listener (ViSQOL)**    To assess perceptual speech quality in an objective manner, we adopt the Virtual Speech Quality Objective Listener (ViSQOL) [15], a signal-based estimator of Mean Opinion Score (MOS) using spectro-temporal similarity.

ViSQOL (in audio mode) assumes a sampling rate of 48 kHz and outputs MOS-LQO scores ranging from 1 to 4.75, with higher scores indicating better perceptual quality. However, since ViSQOL is designed for speech and only supports certain sample rates (e.g., 16 kHz), we additionally employ SigMOS [87] to evaluate speech quality for 48 kHz scenario.

### G.4    Subjective experiment

For the subjective listening test, we evaluate super-resolution performance on generative model outputs from three representative systems: MaskGCT [111], AudioLDM2 [63], and QA-MDT [53], which represent modern speech generation at 24 kHz and music/sound generation at 16 kHz, respectively.

For speech, we randomly select 10 samples from the LibriSpeech-test set [77] as both the reference audio and the text prompt for MaskGCT generation. For music and environmental audio, we sample 10 examples each from the MusicCaps [1] and AudioCaps [40] datasets. To further assess performance on real-world music, we additionally include 10 music segments from the Song-Describer dataset [68]. All audio clips are truncated to 5.12 seconds. For each test case, we present three versions to listeners: the input audio (Generated), the output from AudioSR, and the output from our system (AudioLBM), yielding a total of 40 groups × 3 samples = 120 audio clips for evaluation.

Participants were presented with 40 randomly ordered groups, each containing the three versions described above.They were instructed to rate each sample based on a holistic evaluation of fidelity, clarity, and overall sound quality. Higher scores indicate better perceived quality.

We collected a total of 2400 ratings from 20 participants with diverse backgrounds. We then compute the average ratings across three domains—Speech, Music, and Audio Effects—as shown in Figure 1 in the Introduction. The results show that super-resolution consistently improves the perceptual quality of low-sample-rate audio, confirming the value of upsampling for generative models. Furthermore, our system outperforms AudioSR across all domains, demonstrating the superior audio quality enabled by our approach.

## H    Related works

### H.1    Detailed introduction of audio super-resolution baselines

**Nu-Wave.**    Nu-Wave [49] represents the first attempt at applying diffusion models to waveform-level audio super-resolution. It builds on the WaveNet [106] structure and adopts the DiffWave [44] training paradigm to learn mappings from fixed low-resolution audio to 48 kHz waveform. To support evaluation across various input resolutions, we follow the official implementation[5] and train separate models for 8 kHz, 12 kHz, 16 kHz, and 24 kHz inputs, each for 1.5 million steps.

**Nu-Wave 2.**    Nu-Wave 2 [49] extends Nu-Wave by incorporating two key innovations: Short-Time Fourier Convolution (STFC) and Bandwidth Spectral Feature Transform (BSFT). These additions improve harmonic modeling and allow support for any-to-48 kHz super-resolution. The model also adopts a fast generation scheme using eight predefined non-uniform sampling steps. For fair comparison, we use the pre-trained model made available by the authors[6].

**UDM+.**    UDM+ [120] retains the DiffWave-based architecture but deviates from Nu-Wave in training strategy. It uses unconditional diffusion modeling and injects low-resolution signal guidance at inference through a 50-step uniform sampling process with low-frequency component replacement[67]. Additionally, Manifold Constraint Gradient (MCG) [17] is introduced to better balance low- and high-frequency consistency during generation.

---

[5]`https://github.com/maum-ai/nuwave`
[6]`https://github.com/maum-ai/nuwave2`

**Bridge-SR.** Bridge-SR [54] adapts the Nu-Wave 2 architecture but introduces a key conceptual shift: instead of sampling from a standard Gaussian prior, the model initializes directly from the low-resolution input. This bridge formulation allows high-quality any-to-48 kHz SR. Furthermore, it employs data normalization strategies and frequency-aware loss functions to improve the final reconstruction quality.

**mdctGAN.** mdctGAN [96] targets the instability issues of complex-valued neural networks in the STFT domain by operating in the Modified Discrete Cosine Transform (MDCT) domain. Rather than relying on post-vocoders, mdctGAN uses adversarial learning to generate high-fidelity waveform with phase consistency. We evaluate the model using the authors' official checkpoints[7], covering four different input resolutions.

**NVSR.** NVSR [60] is a neural vocoder-based speech SR system tailored to handle a wide range of upsampling factors and phase reconstruction challenges. It is composed of three main modules: (1) a mel-bandwidth extension network based on ResUNet; (2) a TFGAN-powered [105] neural vocoder; and (3) a post-processing module. We use the official model checkpoint available at [8] for our comparisons.

**AP-BWE.** AP-BWE [65] is the first bandwidth extension system to model the high-frequency phase explicitly. It leverages a GAN-based framework to jointly predict amplitude and phase spectra using a dual-stream CNN, where the two branches interact to reconstruct full-band audio from narrowband speech. Although it does not support arbitrary-resolution inputs, we evaluate the four fixed-resolution models provided by the authors[9].

**AudioSR.** AudioSR [62] is a general-purpose diffusion-based super-resolution model capable of handling speech, music, and general audio inputs. It performs super-resolution in the latent space conditioned on low-resolution inputs and uses a two-stage cascade—consisting of a mel-spectrogram VAE and a vocoder—for waveform reconstruction. Benefiting from strong zero-shot generalization, we evaluate the model using the official implementation[10].

**FrePainter.** Fre-Painter [41] introduces a masked autoencoder (MAE)-based framework for audio super-resolution. It employs an upper-band masking strategy during fine-tuning to simulate low-resolution inputs by masking high-frequency components. Additionally, a mix-ratio masking approach is utilized to enhance robustness across various input sampling rates. The model is pre-trained on large-scale datasets to learn robust speech representations and is fine-tuned jointly with a neural vocoder for waveform reconstruction. We evaluate the model using the official implementation[11].

**FlowHigh.** FLowHigh [122] proposes an audio super-resolution method based on flow matching, aiming to overcome the sampling inefficiency of traditional diffusion models. It adopts an inpainting-style framework similar to AudioSR but constrains the modeling space to Mel-spectrograms. A lightweight two-layer Transformer [107] is used as the velocity predictor. To further improve sampling efficiency, FLowHigh introduces a data-dependent prior instead of a pure Gaussian initialization, allowing for faster flow-based sampling. We evaluate the model using the official implementation[12].

### H.2 Cascaded modeling

### H.2.1 Audio domain

Audio waveform are continuous and high-dimensional [64], training a monolithic network to synthesize full-band signals is notoriously unstable and tends to lose high-frequency detail [16]. Recent work therefore favors cascade architectures in which each stage handles a narrower bandwidth or a simpler representation.

---

[7]https://github.com/neoncloud/mdctGAN

[8]https://github.com/haoheliu/ssr_eval

[9]https://github.com/yxlu-0102/AP-BWE

[10]https://github.com/haoheliu/versatile_audio_super_resolution

[11]https://github.com/FrePainter/code

[12]https://github.com/jjunak-yun/FLowHigh_code

**Text/semantic → audio cascades.** Early systems such as Jukebox [23] map a text prompt through three VQ-VAE tiers—semantic, acoustic and waveform—before decoding; Moûsai [93] reduces this to two latent-diffusion steps. More recent models, including AudioLM, MusicLM, and MusicFlow [9, 1, 81], first generate semantic tokens, then refine them into acoustic tokens that a vocoder converts to the final audio. InspireMusic [123] pairs a transformer LM with a flow-based super-resolution decoder for long-form, high-resolution music, while YuE [121] uses an LLM-based lyric encoder followed by a lightweight vocoder, forming a two-stage lyric-to-song pipeline.

**Super-resolution cascades.** On the SR side, models improve fidelity by letting each block focus on a limited frequency band: progressive up-sampling GANs (PU-GAN) [16], coarse-to-fine phase extensions such as MS-BWE [66], and multi-band diffusion that processes sub-bands in parallel [92]. The work most closely related to ours, Noise-to-Music [36], cascades two diffusion models (3.2 kHz → 16 kHz) and uses waveform blurring plus stochastic low-pass resampling to perform cascading augmentation.

### H.2.2 Vision domain

A similar multi-stage processing trend is evident in the vision domain. Cascade-resolution training progressively feeds the same network with increasingly finer inputs—as seen in models like PixArt-Σ[12] and SANA[116]—enabling a single model to scale seamlessly to generate high-resolution images.

**Text-to-Image cascades.** Modern text-to-image generation pipelines often employ cascaded architectures, where separate diffusion or autoregressive models operate sequentially at increasing spatial resolutions. Early examples include CDM [35], DALL·E 2 [83], Imagen [90], and Relay-Diffusion [104]. More recent advances, such as CogView 3 [127], Matryoshka Diffusion Model [31], and FMboost [95], incorporate hybrid approaches combining diffusion models or flow models. Pipelines like SDXL [80] enhance the image quality of Stable Diffusion [88] through explicitly incorporating multi-resolution enhancement and refinement stages.

**Super-resolution cascades.** Super-resolution systems built on diffusion models also adopt cascaded frameworks. Methods such as Inf-DiT [119] and SR3 [91] perform iterative refinement, while Pixel-Space Laplacian Diffusion [2] leverage successive Laplacian pyramid-based enhancements. Together, these works confirm that allocating separate generators to coarse layout and fine detail is crucial for stable training, fast sampling, and high-fidelity ultra-resolution imagery.

## I   Qualitative results

In this section, we present two subsections of case studies. The first subsection compares the super-resolution performance of bridge models trained in different representation spaces. The second focuses on cherry-picked samples from the demo page of $A^2SB$ [45], which provides a strong baseline for comparison.

### I.1   Modeling on different audio spaces

In this section, we compare three bridge models: (1) following Bridge-SR [54], we train a 10.6M-parameter network based on a large WavNet [106] architecture on our dataset; (2) following NVIDIA-NeMo [46][13], we construct a bridge model directly on the STFT spectrogram, treating each time frame as a token and modeling it with a DiT (Diffusion Transformer [4]) architecture of 0.3B, also trained on our dataset; and (3) our proposed *any-to-48 kHz* AudioLBM system. As shown in Figure 12, waveform-based bridge models perform reasonably well on speech data but remain constrained by limited model capacity. When applied to more diverse content such as music and sound effects, these models fail to generalize, resulting in blurred high-frequency components and weak spectral energy. In contrast, STFT-based models can roughly capture the relationship between low and high frequencies; however, the high-frequency details remain overly smooth, and harmonic structures are not well preserved—often leading to visible spectral discontinuities. Our proposed

---

[13]`https://github.com/NVIDIA/NeMo`

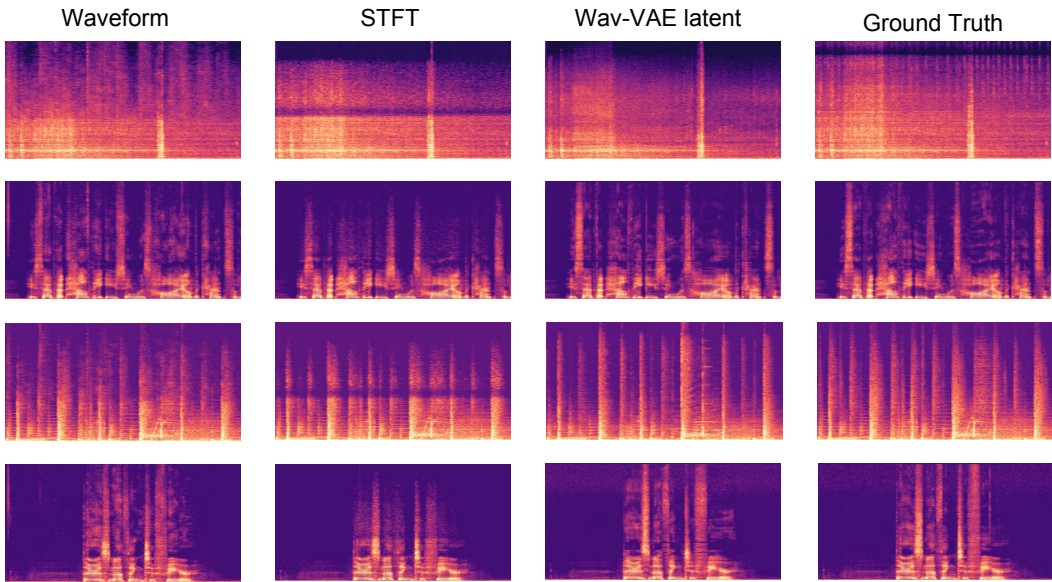

Figure 12: Super-resolution results for bridge models on different modeling spaces.

AudioLBM demonstrates better generalization across all domains, yielding richer spectral detail and more consistent energy distribution across the frequency spectrum.

## I.2 Comparison with cherry-picked samples vs. our non-cherry-picked results

In this section, we present a qualitative comparison across five representative super-resolution baselines on the 8 kHz to 48 kHz task: AudioSR [62], A$^2$SB [45], re-implemented 48 kHz version of Audit, CQTDiff [72] form from A$^2$SB [41], and our proposed method. As shown in Figure 13,

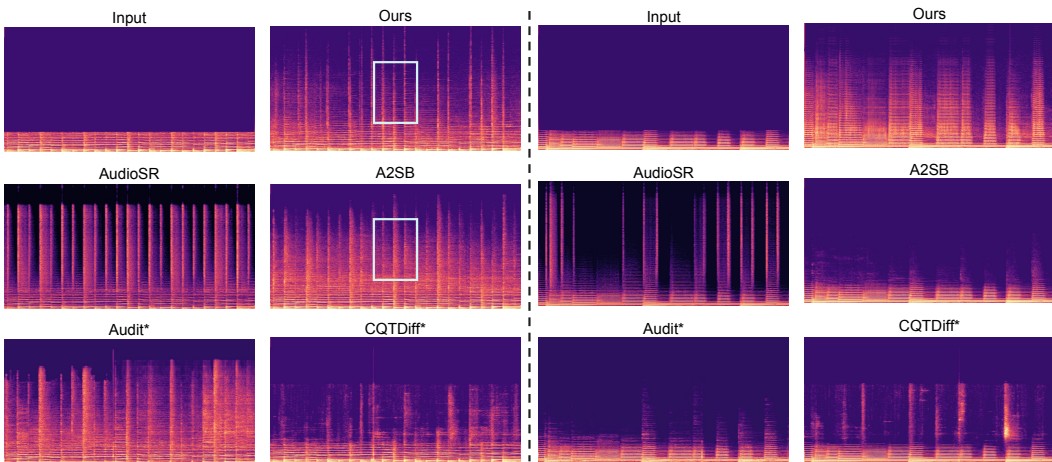

Figure 13: Qualitative comparison from A$^2$SB's demo page.

our method clearly outperforms other latent-representation-based approaches such as AudioSR and Audit, producing more complete, harmonic-rich, and structurally coherent high-frequency content. In contrast, direct data-space methods like A$^2$SB and CQTDiff tend to generate overly smooth outputs with blurred high-frequency regions and fail to reconstruct clear harmonic patterns—especially visible in the blue box in the leftmost example.

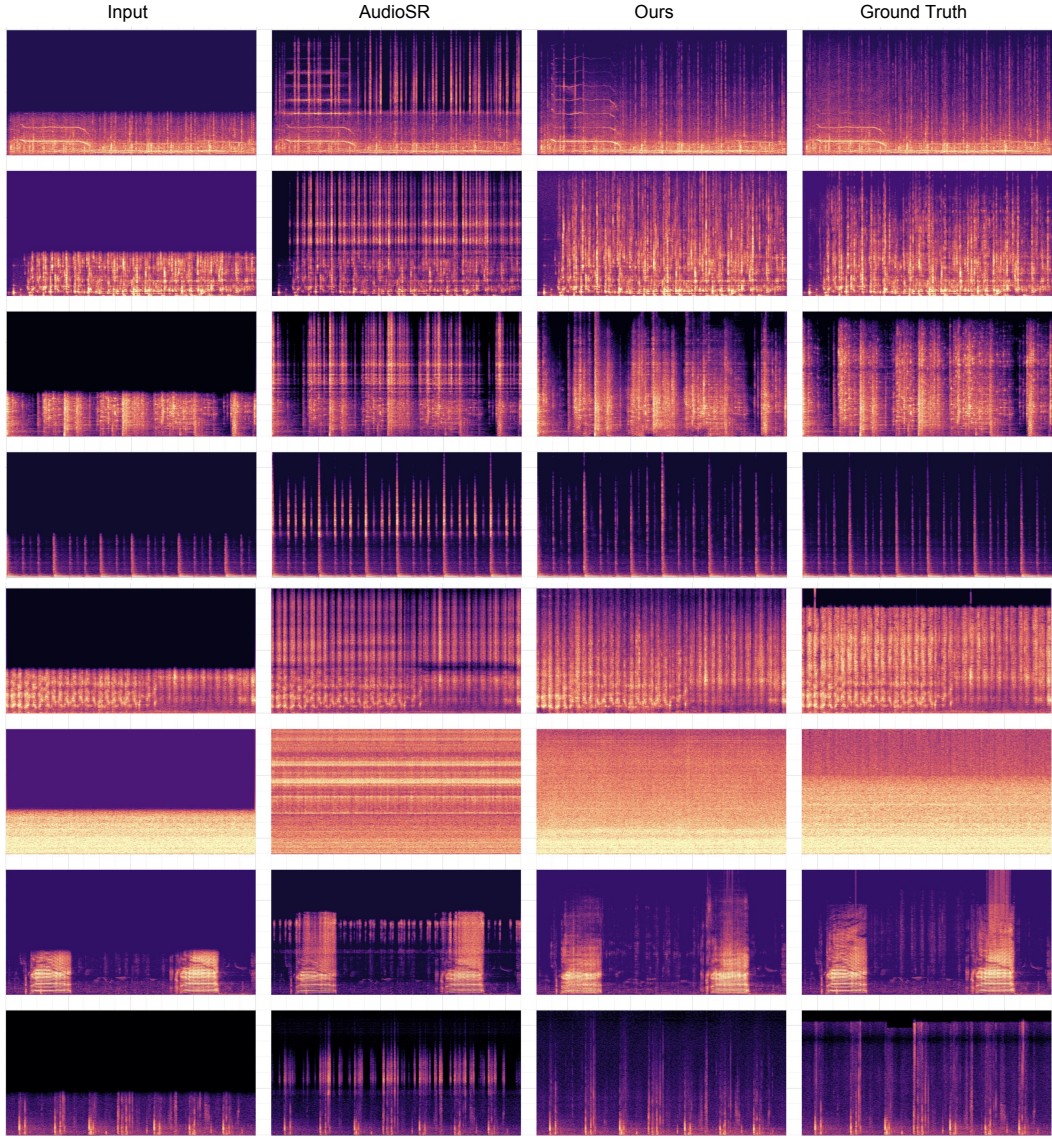

Figure 14: Additional comparison with AudioSR.

## I.3 Additional Comparison

As shown in Figure 14, we further compare eight 16 kHz to 48 kHz super-resolution samples between our method and AudioSR. It can be observed that AudioSR, which performs spectral completion in the latent space, often produces excessively strong high-frequency components or misalignments between high- and low-frequency regions. This suggests a limited ability to leverage the low-frequency cues present in the input.

Additionally, due to the use of multi-stage cascading compression network in AudioSR, fine spectral detail is often smoothed out, resulting in overly blurred outputs. In contrast, our method aligns more closely with the ground truth and maintains strong coherence across the full frequency range. Moreover, thanks to the *any-to-any* training paradigm, our system can consistently generate full-bandwidth 48 kHz outputs, whereas AudioSR occasionally fails to fully reconstruct the high-frequency spectrum.

