# OpenReview forum: "Audio Super-Resolution with Latent Bridge Models"
_NeurIPS.cc/2025/Conference — NeurIPS 2025 poster_

### Official Review · Reviewer_X2LP · 2025-06-26

**Clarity:** 4
**Significance:** 4
**Originality:** 3
**Rating:** 5
**Confidence:** 5

**Summary:**

The paper proposes to tackle audio super resolution using Latent Bridge Models (LBMs).
The authors propose to condition the LBM (bridging between low resolution input to high resolution output) on the input and target frame rates, therefore enabling any-to-any upsampling.
They also propose a super resolution model for very high sampling rates (up to 192kHz) using cascading LBMs and propose methods to mitigate the errors arising with multi-stage generation.
There's an accompanying website in which they compare their approach with AudioSR, a similar approach based on conditional Latent Diffusion Models.

**Questions:**

- Will the the models & code will be publicly available and reproducible?

**Ethical Concerns:**

["NO or VERY MINOR ethics concerns only"]

**Limitations:**

Ideas in this paper are fairly standard so the main value of this work is in the models and their implication in the audio community.

**Quality:**

4

**Strengths And Weaknesses:**

Strengths:
- Consider very high sampling rates (up to 192kHz) which is quite rare in the domain
- state-of-the art performances. Clear improvements over audioSR. Examples on the accompanying website corroborate the good results on the metrics.
- The modeling choices, if quite standard, are well-motivated (choice of LBM instead of LDM), conditioning on framerate to leverage more data, augmentations when cascading; which leads to a convincing ablation study.
- Well-written and extensive comparison
- Good justification about why 192kHz is sometimes required for professional use
- If the model is released, this is an excellent contribution that may be quite impactful, as the community working on audio generative modeling could work at lower sampling rates without sacrificing audio quality.

Weaknesses:
- not clear if the models & code will be publicly available.
- Lacking details on the high-res VAE.
- Maybe the sampling time can be stated explicitly so that practitioners may assess if this methods

---

> ### Author Rebuttal · Authors · 2025-07-30
>
> We thank the reviewer for the valuable and inspiring comments.
>
> ### **Main Comments:**
> ---
>
> #### **W1 and Q1: If models & code will be publicly available**
> We appreciate the reviewer's emphasis on reproducibility, which we also consider highly important. We will make our best efforts to release more implementation details and resources to support further research in the audio community, e.g., opening the inference implementation in the first step.
>
> #### **W2: Lacking details on the high-res VAE**
> We agree that the high-resolution VAE plays a crucial role in our framework.
>
> In fact, we have provided extensive details in **Appendix**, and we evaluate both the **reconstruction** and corresponding **generation** performance of our VAE by training for **200K steps** on each configuration. We train each ablated VAE configuration on our full speech corpus (details in **Appendix G**) for 200K steps, and pair it with a corresponding Latent Bridge Model (LBM) trained for 150K steps. All evaluations are conducted on the VCTK-test set. We kindly refer the reviewer to the following sections for more specifics:
>
> - In **Appendix B.1**, we outline the VAE architecture and loss formulation, which closely follow the training structure of **Stable-Audio-Open [1]** and **ETTA [2]**. Specifically, we adopt a Oobleck[3]-based compression network with DAC[4]-based discriminator.
> - In **Appendix B.1.1**, we conduct a systematic study of **compression ratios** for super-resolution, ultimately selecting a **512× compression** from waveform $ \mathbf{x} \in \mathbb{R}^L $ to latent $ \mathbf{z} \in \mathbb{R}^{L/512 \times 64} $, and we find that a time-axis compression rate of 512 achieves the best trade-off between efficiency and performance.
> - **Appendix B.1.2** investigates the effect of **KL divergence weight**, and includes both quantitative analysis and interpretability discussion. We apply a fixed scaling factor s introduced in Section 3.2, line 166 as 0.25 and KL divergence as 0 (where our VAE degrades into an AutoEncoder).
> - **Appendix B.1.3** compares our VAE reconstruction quality against recent open-source baselines on 48 kHz audio.
> - **Appendix B.2** details the **training procedures for high-resolution VAE models** at 96 kHz and 192 kHz, including parameter choices and convergence behavior.
>
> ##### 96Audio dataset
>
> | Step | Phase    | Model                      | frame rate    | SSIM ↑ | LSD ↓  | LSD-LF ↓ | LSD-HF ↓ |
> |------|----------|----------------------------|-------|--------|--------|-----------|-----------|
> | 100w | pretrain | 48 kHz-vae                 | 50Hz  | 0.815  | 0.808  | 0.815     | 0.774     |
> | 100w | pretrain | 96 kHz-vae                 | 50Hz  | 0.818  | 0.798  | 0.803     | 0.772     |
> | 100w | pretrain | 48 kHz-vae                 | 100Hz | 0.840  | 0.811  | 0.821     | 0.760     |
> | 100w | pretrain | 96 kHz-vae                 | 100Hz | 0.841  | 0.731  | 0.747     | 0.698     |
> | 20w  | finetune | finetune from 48 kHz-vae    | 100Hz | **0.850**  | **0.709**  | **0.703**     | **0.692**     |
>
>
> ##### 192Audio dataset
>
> | Step | Phase    | Model                      | frame rate    | SSIM ↑ | LSD ↓  | LSD-LF ↓ | LSD-HF ↓ |
> |------|----------|----------------------------|-------|--------|--------|-----------|-----------|
> | 100w | pretrain | 192 kHz-vae                | 100Hz | 0.868  | 0.736  | 0.643     | 0.752     |
> | 100w | pretrain | 96 kHz-vae                 | 100Hz | 0.862  | 0.747  | 0.646     | 0.763     |
> | 100w | pretrain | 48 kHz-vae                 | 100Hz | 0.863  | 0.750  | 0.692     | 0.748     |
> | 20w  | finetune | finetune from 96 kHz-vae   | 100Hz | 0.866  | 0.722  | 0.630     | 0.740     |
> | 20w  | finetune | finetune from 48 kHz-vae   | 100Hz | **0.871**  | **0.713**  | **0.603**     | **0.737**     |
>
> For the 96 kHz and 192 kHz VAEs, as shown in **Appendix B.2**, we adopt the same network architecture and training procedure as the 48 kHz VAE. Moreover, the higher-resolution VAEs are fine-tuned from the lower-resolution ones to alleviate the scarcity of ultra high-resolution training data.
> We will make this clearer in the final version.
>
> ---
>
> #### **W3: Sampling time can be stated explicitly**
>
> The inference efficiency is indeed a key factor for the deployment of high-resolution audio generation models.
>
> To address this, we present below the real-time factor (RTF) and number of function evaluations (NFE) of our method and several representative baselines on a single NVIDIA-A800, all evaluated under the 48 kHz setting:
>
> | Method                   | Modeling Space | Model             | RTF ↓ | NFE |
> | ------------------------ | -------------- | ----------------------- | ----- | --- |
> | **Ours (any → 48 kHz)**   | Wav-VAE        | Bridge                  | 0.369 | 50  |
> | **Ours (any → 96 kHz)**  | Wav-VAE        | Cascaded Bridges        | 0.695 | 100 |
> | **Ours (any → 192 kHz)**  | Wav-VAE        | Cascaded Bridges        | 1.351 | 150 |
> | Bridge-SR [5] 48 kHz     | Waveform       | Bridge                  | 1.670 | 50  |
> | UDM+ [6]  48 kHz         | Waveform       | Unconditional Diffusion | 2.320 | 100 |
> | AudioSR [7]     48 kHz   | Mel-VAE        | Conditional Diffusion    | 0.948 | 50  |
> | Fre-painter [8]   48 kHz | Mel            | GAN                     | 0.009 | 1   |
> | NVSR [9]  48 kHz         | Mel            | GAN                     | 0.033 | 1   |
>
> While our method may not match the inference speed of GAN-based models like [8] and [9], it offers a substantial speedup over other **iterative-based** models, including Bridge-SR [5], UDM+ [6], and previous SoTA AudioSR [7] (RTF 0.369 vs. 0.948).
>
> When scaling the sampling target to higher sampling rates (e.g., 192 kHz), it naturally leads to slower inference speeds beacuse of the cascade paradigm. Especially, we mitigate the impact by adopting a lighter network architecture, as shown in Appendix G.2, so the inference speed does not increase linearly. As shown, even upsampling to 192 kHz, it is even faster than the 48 kHz waveform-domain bridge system [5] (RTF 1.351 vs. 1.670).
>
> We hope our reply can address your concerns!
>
> ---
>
> [1] Stable Audio Open. ICASSP, 2025
>
> [2] ETTA: Elucidating the Design Space of Text-to-Audio Models. ICML, 2025
>
> [3] Long-form music generation with latent diffusion. ISMIR, 2024
>
> [4] High-fidelity audio compression with improved rvqgan. NeurIPS, 2023
>
> [5] Bridge-SR: Schrödinger bridge for efficient sr. ICASSP, 2025
>
> [6] Conditioning and sampling in variational diffusion models for speech super-resolution. ICASSP, 2023
>
> [7] AudioSR: Versatile audio super-resolution at scale. ICASSP, 2024
>
> [8] Audio super-resolution with robust speech representation learning of masked autoencoder. TASLP, 2024
>
> [9] Neural vocoder is all you need for speech super-resolution. Interspeech, 2022

---

### Official Review · Reviewer_UGNU · 2025-07-02

**Clarity:** 3
**Significance:** 2
**Originality:** 2
**Rating:** 4
**Confidence:** 3

**Summary:**

This paper tackles the long-standing problem of sub-optimal audio super-resolution caused by uninformative generative priors. The authors propose an Audio Latent-Bridge Model (Audio-LBM) that first compresses a waveform into a continuous latent space and then learns a generative bridge from low-resolution (LR) to high-resolution (HR) latents. To reduce data scarcity at high sampling rates, they introduce a frequency-aware LBM that conditions on both LR and HR cutoff frequencies, enabling any-to-any up-sampling within a single model. A cascaded LBM configuration is further adopted to reach 48 kHz -> 192 kHz. To limit error accumulation across stages, two prior-augmentation techniques are applied: (i) waveform-domain degradation that removes fine-grained details near the Nyquist boundary, and ((ii) latent-domain dynamic Gaussian smoothing. Comprehensive experiments on speech, general audio, and music benchmarks show state-of-the-art results for any-to-48 kHz and, for the first time, strong baselines for any-to-192 kHz.

**Questions:**

- The paper states: “We directly compress the audio waveform into a continuous latent representation, where the latent of the LR waveform provides instructive information for the HR latent and avoids the discarded regions in previous works [20, 22].”
Could you supply concrete evidence for this claim? There are many ways to map a waveform into a continuous latent space, and it is not obvious that the chosen compression actually preserves the information needed to reconstruct HR audio. Any experiments comparing alternative latent encoders (e.g., raw-waveform or mel-spectrogram latents) would help substantiate the point.

- You argue that conditioning on both input and target cutoff frequencies allows the model to learn an any-to-any up-sampling process and boosts 48 kHz performance. The ablations indeed show a gain, but why does training with lower-band samples help when the final target is fixed at 48 kHz? A deeper analysis—attention visualizations, latent interpolations, or other diagnostics—would clarify the mechanism. Right now the paper attributes the gain mainly to larger data scale, which feels insufficient.

- The ablation study covers dataset filtering and frequency awareness, but it does not isolate the benefit of the latent-bridge formulation itself. To make the argument stronger, please add direct comparisons with (i) latent diffusion models trained on the same latents and (ii) bridge models that operate on raw waveform or spectral representations. Such head-to-head results would clarify how much improvement truly comes from adopting a latent bridge versus alternative generative priors.

**Ethical Concerns:**

["NO or VERY MINOR ethics concerns only"]

**Final Justification:**

I have thoroughly read the authors’ reviews and rebuttal, and many of my concerns have been addressed. While this paper is technically solid, I believe there are additional aspects that need to be addressed for its academic contribution to be applied to a broader domain, so I intend to keep my score unchanged.

**Limitations:**

yes

**Paper Formatting Concerns:**

No significant formatting issues were found.

**Quality:**

3

**Strengths And Weaknesses:**

Strengths:
The paper presents a robust model that delivers consistently strong results for any-to-48 kHz audio super-resolution across speech, music, and general audio, clearly outperforming prior work. It backs this claim with extensive experiments and ablation studies that reveal how each proposed component contributes to the final quality. The released audio samples sound noticeably better than baselines, and the framework is scaled to 96 kHz and 192 kHz—rates that matter in industry yet are rarely explored in academia. Importantly, the authors do more than stack models: they introduce cascade-error-mitigation techniques that meaningfully curb quality degradation between stages.

Weaknesses:
While the system works well, the paper offers little scientific insight into why the latent-bridge approach yields more informative priors than diffusion or GAN alternatives. Discussion of broader applicability is limited to a brief “future work” remark, leaving unanswered how the method might generalize to other restoration tasks or modalities. A deeper analysis of the underlying mechanisms—and clearer guidance on extending the technique—would strengthen the contribution.

---

> ### Author Rebuttal · Authors · 2025-07-30
>
> We thank the reviewer for the valuable comments. We provide a point-to-point response as follows.
>
> ### **Main Comments:**
> ---
>
> #### **W1: little scientific insight into why the latent-bridge approach yields more informative priors than diffusion or GAN alternatives**
> Thank you for pointing this out. We respectfully clarify the scientific motivation behind using the latent-bridge approach, which fundamentally lies in how it redefines the **prior distribution** of the generative process in a way that is more SR task-aligned and informative than the standard Gaussian prior used in conventional diffusion models.
>
> In  diffusion models, the trajectory is defined as $
> \mathbf{z}_ {t,diff} = \alpha(t) \cdot \mathbf{z}_ 0 + \beta(t) \cdot \boldsymbol{\epsilon}, \boldsymbol{\epsilon} \sim \mathcal{N}(0, \mathbf{I}),
> $ although the LR input is injected as a condition, the generative process itself starts from an uninformative noise prior (typically $\mathbf{z}_T \sim \mathcal{N}(0, \mathbf{I})$). This **noise-to-data** generation imposes a heavy burden on the model: it must learn to reconstruct the HR target solely from random noise, while separately relying on the condition for guidance. As a result, the **prior distribution is misaligned with the actual LR-to-HR task**, leading to inefficiency and potential degradation in performance.
>
> In contrast, our latent-bridge model formulate:
> $
> \mathbf{z}_ {t,bridge} = \frac{\alpha_ t  \bar{\sigma}_ t^2}{\sigma_ 1^2} \, \mathbf{z}_ 0+ \frac{\bar{\alpha}_ t \, \sigma_ t^2}{\sigma_ 1^2}  \mathbf{z}_ {T,bridge} + \frac{\alpha_ t  \bar{\sigma}_ t  \sigma_ t}{\sigma_ 1} \\boldsymbol{\epsilon}, \boldsymbol{\epsilon} \sim \mathcal{N}(0, \mathbf{I})
> $ and introduces a **data-informed** prior, where the sampling process starts from an LR latent representation that is already informative about the HR target. This establishes a data-to-data generation trajectory.
>
> Moreover, compared to GANs, diffusion and bridge models adopt **iterative sampling**, which enables the model to better capture fine-grained details, potentially leading to improved super-resolution quality.
>
> #### **W2: How the method might generalize to other restoration tasks or modalities**
>
> Your question is sincerely appreciated. We totally agree that more discussion of the underlying mechanisms can strengthen our contribution. Also, we provide our insights for extending this method to other restoration tasks as follows.
>
> **More discussion for generalizable training**: from our perspective, bridge models are advantageous in the tasks where the condition information is instructive to the generation target. However, without a compression network, directly designing a bridge model in the waveform space or the STFT representations may increase the burden of generative models, as the redundant information in the data space has to be accurately modelled with a single generative model.
>
> In our method, we transform the **data-to-data** generation process into a **latent-to-latent** one with a waveform compression network, which preserves the advantages of bridge models on exploiting informative prior while enabling more compact and generalizable training, allowing our AudioLBM to improve the quality across speech, audio, and music samples as shown in the subjective quality results in our **Figure 1**.
>
> **Extending to other restoration tasks**: we think our method will be indicative to the design of bridge models on speech restoration tasks, especially the general speech restoration systems, as these tasks also require generalizable training and has indicative observation for the generation target. However, developing bridge models for these tasks should further consider their task-specific evaluation metrics, and then propose the innovative techniques suitable for these tasks.
>
> #### **Q1: The paper states: “We directly compress the audio waveform into a continuous latent representation, where the latent of the LR waveform provides instructive information for the HR latent and avoids the discarded regions in previous works [20, 22].” Could you supply concrete evidence for this claim? There are many ways to map a waveform into a continuous latent space, and it is not obvious that the chosen compression actually preserves the information needed to reconstruct HR audio. Any experiments comparing alternative latent encoders (e.g., raw-waveform or mel-spectrogram latents) would help substantiate the point.**
>
> Thank you for your valuable suggestion. We agree that the connection between compression strategy and the preservation of HR information should be better articulated.
>
> However, we would like to respectfully clarify that the term "discarded regions", also referred to as "area removal", is **borrowed from** the recently proposed STFT-domain bridge model $A^2SB$ [1]. It describes the phenomenon in which high-frequency components are **effectively discarded** when using STFT or mel-spectrogram representations[2]. As the input has a lower effective frequency band, the area removal problem becomes more pronounced, making the inpainting task **increasingly challenging**.
>
> We would also respectfully clarify that our discussion concerns the conceptual difference between **"prior"** and **"condition"** in the context of probabilistic generative processes, which respectively correspond to the **boundary distribution of generative process** (also the advantage of our bridge model compared to diffusion model) and the **auxiliary indicative input of network network**.
>
> To substantiate our choice, we conduct additional experiments **(see Q3)** comparing waveform-based latent compression with alternatives using STFT and mel-spectrogram representations. These results confirm that our waveform latent preserves richer frequency content and leads to superior super-resolution performance.
>
> We will make these motivations and comparisons clearer in the final version. Thank you again for helping us improve the clarity of our work, we will clarify this with more explanations in the final version.
>
>
> ---
>
> #### **Q2: Deeper analysis of the *any-to-any* process**
>
> We would like to clarify that, by defining these two conditions, our bridge models can gain a clearer understanding of the frequency boundary between the LR prior and the HR target in the bridge upsampling process. In other words, the generative process is explicitly conditioned on these two additional pieces of start and end points, enabling the model to jointly learn multiple super-resolution tasks during training. Hence, this finer-grained guidance allows the model to make more effective use of all available data (rather than relying solely on full-band inputs), and can generate desired full-band result even trained with waveforms suffered from limited effective frequency ranges, ultimately improving super-resolution performance.
>
> ---
>
> #### **Q3: Add direct comparisons with (i) latent diffusion models trained on the same latents and (ii) bridge models that operate on raw waveform or spectral representations. Such head-to-head results would clarify how much improvement truly comes from adopting a latent bridge versus alternative generative priors.**
>
> Thank you for the valuable suggestion. We agree that isolating the contribution of the latent-bridge formulation is important for a fair evaluation.
>
> To address this, we have conducted additional experiments using the same speech datasets (OpenSLR [3], Expresso [4], EARS Dataset [5], and VCTK-Train [6]) for training all models, and evaluated them on the same VCTK-Test set.
>
> We compare the following variants without any additional modules, each model is trained with 500K steps on the any-to-48 kHz setting and tested on 8-to-48 kHz SR setting with 50-step sampling:
>
> - Latent Diffusion (on wav-vae latents)
> - Latent Rectified Flow (on wav-vae latents)
> - Bridge-STFT (Based on the Nemo[5] framework)
> - Bridge-Waveform (Based on Bridge-SR[6])
> - Latent Bridge (Ours)
>
> | Method                   | Modeling Space   | SSIM ↑ | LSD ↓| LSD-LF ↓| LSD-HF ↓|SigMOS[9] ↑|
> |--------------------------|----------------|--------|-------|-------|-------|------|
> | Diffusion[7] | Mel-vae latent| 0.809 | 0.940 |0.486|0.994|2.846 |
> | Rectified Flow[8] | Mel | 0.784 | 0.816 |0.194|0.889|2.792 |
> | ***Bridge***         | Complex STFT           |  0.809  | 1.295 | 0.414|1.401 |2.951|
> | ***Bridge***        | Raw waveform       |  0.660  | 1.037 |**0.184**|1.101| 2.896 |
> | Rectified Flow    | ***Ours wav-vae latent***     |  0.880    | 0.751 | 0.793|0.722| 2.892|
> | Diffusion         | ***Ours wav-vae latent***     |   0.879   | 0.758 | 0.806|0.728 |2.741 |
> | ***Bridge (Ours)*** | ***Ours wav-vae latent*** | **0.907** |  **0.742** | 0.708 | **0.712** | **3.095** |
>
> As shown above, our latent-bridge model achieves both higher SSIM, LSD (objective quality) and SigMOS (subjective quality) score, supporting that the **latent-domain bridge formulation contributes significantly** to final performance, even when controlling for model architecture and training data.
> We also provide some corresponding **visualizations** in **Appendix I.1** to make this clearer.
>
> ---
>
> [1] A2SB: Audio-to-Audio Schrodinger Bridges. arXiv preprint arXiv:2501.11311 (2025).
>
> [2] Time-frequency Networks for Audio Super-resolution. ICASSP, 2018
>
> [3] Crowd-Sourced Speech Corpora for Javanese, Sundanese, Sinhala, Nepali, and Bangladeshi Bengali. SLTU, 2018.
>
> [4] Expresso: A benchmark and analysis of discrete expressive speech resynthesis. Interspeech, 2023
>
> [5] Speech enhancement and dereverberation with diffusion-based generative models. TASLP, 2023
>
> [6] Expressive neural voice cloning. ACML, 2021.
>
> [7] AudioSR: Versatile audio super-resolution at scale. ICASSP, 2024
>
> [8] Flowhigh: Towards efficient and high-quality audio super-resolution with single-step flow matching. ICASSP, 2025
>
> [9] ICASSP 2024 speech signal improvement challenge. IEEE Open Journal of Signal Processing, 2025
>
> ---

---

> > ### Comment · Reviewer_UGNU · 2025-08-06
> >
> > I would like to thank the authors for their detailed responses to my comments and questions. The authors have provided sufficient clarification, and I especially appreciate their effort in conducting additional experiments for Q3 within a short time to strengthen their claims. The concerns I had during the initial review have been well addressed, and I believe this is a technically solid piece of work. Nevertheless, I find the overall contribution to the academic community to be somewhat narrow in scope, and therefore I will maintain my original score. Thank you again for the excellent work and thoughtful responses.

---

> > > ### Author Response · Authors · 2025-08-06
> > >
> > > We sincerely thank the reviewer for their thoughtful response and kind recognition. We truly appreciate your encouraging feedback. Looking ahead, we are excited to explore the potential opportunities for applying our proposed method to other domains. Once again, thank you for your valuable time and effort in reviewing our work.

---

### Official Review · Reviewer_sHec · 2025-07-03

**Clarity:** 4
**Significance:** 4
**Originality:** 4
**Rating:** 5
**Confidence:** 4

**Summary:**

This paper presents a method to increase the bandwidth of an audio signal ("super-resolution") to a target frequency of sampling in an "any-frequency-to-any" fashion. The method operates by (conditioned) Shrödinger-Bridge diffusion over a latent-space representation obtained by a VAE on time-domain signals. An approach to cascade such models to reach higher frequencies that have no been considered in the literature before (96 kHz and 192 kHz) is also presented. A detailed experimental study including extensive comparisons to the state of the art on multiple datasets over objective and perceptual metric, and ablation study and a case study are presented.

**Questions:**

Questions and Suggestions:
-  L11: "the prior and target frequency". Please specify: "the prior and target *frequency of sampling*" or *Nyquist frequency* or *max frequency* or *cut-off frequency* (otherwise not clear).
- Please clarify in the caption of Fig. 2 what the 3 parts of the figure represent. If I understood right, the middle part is NOT the proposed approach but the competing approach AudioSR, while only the bottom part is AudioLBM (and I am not sure exactly if the top part represents one, or the two, or both, nor what it is meant to convey).
- To justify upsampling to 96 kHz or above (which is not obviously useful a priori, at least to me), the authors first write "allows flexibility in audio post-production" at line 60 but provide no references, then "provides further benefits" at line 85 and cite [33,2,3,57] and then "provides engineering advantages and post-processing flexibility" and cite a different paper [51]. Please harmonize this by providing all of the relevant references every time.
- L113: it would be good to specify here whether the VAE is trained on LR or on HR waveform (or both). I didn't find the information elsewhere in the paper.
- L142: "the diffusion-based audio upsampling" -> what diffusion-based audio upsampling exactly?
- L145: Could you please clarify what is meant by "which has been aligned with" here?
- L155: "while it has already provided", I don't understand this sentence. What does "it" refer to here?
- In the authors' framework, both HR and LR signals have the same length L. But it is not clear in practice how, given a shorter LR signal sampled at a lower rate, its size is increased to L. The authors only write at L231 "all data are resampled to the corresponding target sampling rate". But such process can be prone to artifacts that may impact the results. Please clarify how this was handled.
- L164: please define the acronym DiT before and explain the corresponding model, for example here or when you introduce the training objective (2).
- L166: why plural: "bridge models" ? are several models trained?
- It would be nice to include some idea of the inference computational cost of the method (especially for the cascaded version, which the authors say is costly).
- Typos
    - L38, has->have
    - L79: limited in specific domains -> to specific domains
    - L170 - low-pass filtered
    - L235 and L238: please improve/avoid the dual use of commas, eg (1,000,20,000), which makes it hard to read. (eg (1, 20) kHz instead)
    - L291: "sorely" -> solely
    - Beware of putting curled bracket around accronyms in bibtex references to make them appear correctly in the bibliography, e.g., Audiosr -> AudioSR

**Ethical Concerns:**

["NO or VERY MINOR ethics concerns only"]

**Final Justification:**

I thank the authors for their clear and detailed responses. I stand by my recommendation of 5: Accept, which I think is well-calibrated for this submission.

**Limitations:**

yes (end of conclusion).

**Paper Formatting Concerns:**

no problem.

**Quality:**

4

**Strengths And Weaknesses:**

Strengths:
- This is generally high-quality research, with a clear and reproducible presentation of the work. The reproducibility is strengthened by the addition of 24 pages of implementation details in appendix.
- The topic is interesting and relevant
- The intro and related work section are very well done and well written.The literature review is of high quality and thorough, with 70 citations.
-  the new ideas of using latent bridge diffusion and of conditioning by the input and target frequencies are sound and well executed.
- The techniques presented for cascading models are smart and original
- The experiments are of high quality, including detailed comparison across many references, datasets and relevant metrics, an ablation study, and a qualitative case study.
- The proposed approach clearly outperforms the state of the art over both objective and perceptual metrics across all test datasets.

Weaknesses:
- The motivation behind  upsampling audio signals to 96kHz or above is not clear. Features at that scales are clearly inaudible by the human auditory system. The authors mention "allows flexibility in audio post-production" -> is there some reference/evidence for this?
- A few points should be clarified in the presentation of the method (see questions).

---

> ### Author Rebuttal · Authors · 2025-07-30
>
> We thank the reviewer for the valuable comments. We provide a point-to-point response as follows.
>
> ### **Main Comments:**
> ---
>
> #### **W1 and Q3: Motivation behind upsampling beyound 48 kHz and harmonize citation by providing all of the relevant references**
>
> We will rigorously verify and manage citations regarding the role of ultra‑high sampling rate audio. Specifically, we will address our motivation in **mastering [1], post‑production [2,3], spatial audio[4], and immersive content creation[5]**.
>
> As [1] shows, adopting 96 kHz permits gentler anti‑alias filters and greater headroom in the final master, albeit with higher data‑rate costs; [2] identifies near ultrasonic aliasing that intermodulates during editing and playback, advocating higher sample rates to suppress this distortion in production chains; [3] presents a meta‑analysis demonstrating a small but statistically significant perceptual advantage for high‑resolution audio under controlled listening conditions, supporting its use in critical post‑production monitoring; [4] provides psychophysical evidence that high‑frequency cues that are better preserved at ultra high sampling rates are vital for accurate spatial localization and [5] outlines coding strategies that maintain full‑bandwidth fidelity, a prerequisite for convincing, immersive playback formats.
>
> #### **Q1: Specify: "the prior and target frequency of sampling"**
>
> The term "frequency" here refers to the **effective cut-off frequency**, as defined as $ \text{SR}_ {x_ {\text{HR}}} $ in Section 3.2 and explained as $ f_ {\text{eff}} $ in Appendix A.1. This represents the **maximum frequency containing meaningful content** in the audio signal and is **independent of** the waveform’s sampling rate.
>
> In our training setup, the LR signal is filtered from the HR waveform, sharing the same length. The prior and target frequencies we mention correspond to the **effective cut-off frequencies of the LR and HR waveforms**, respectively. These values are included as conditioning signals to enable the any-to-any super-resolution process described in Section 3.2.
>
> We will add more explanations to clarify this in the final version.
>
> ---
>
> #### **Q2: Clarify the caption of Fig. 2**
>
> Thank you for pointing this out. To clarify:
>
> - The **top part** of Figure 2 illustrates how the **low-resolution waveform is simulated during training** using low-pass filtering.
> - The **middle part** corresponds to the **baseline method AudioSR**, which synthesizes high-resolution content from Gaussian noise.
> - The **bottom part** presents our proposed method, **AudioLBM**, which defines a bridge-based trajectory between low- and high-resolution latent variables.
>
> We will modify our caption of Figure 2 in the final version.
>
> ---
>
> #### **Q4: Specify whether the VAE is trained on LR or on HR waveform (or both)**
>
> Thank you for your valuable suggestion. Our 48 kHz VAE is trained **solely on the original 48 kHz (HR) resolution waveforms**,. We also experimented with supplementing the training data using low-pass-filtered (i.e., **LR**) versions of the same signals, but we did not observe improvements in reconstruction or super-resolution quality.
>
> To address data scarcity at higher resolutions (96 kHz and 192 kHz), we adopt a staged training strategy, as described in **Appendix B.2**: we initialize the VAE with the weights from the lower sampling rate model and then **fine-tune** on the corresponding **HR** waveform data at each higher sampling rate. Please kindly visit our **Appendix B** for more details.
>
> We will add more explanations on this point in the final version.
>
> ---
>
> #### **Q5: L142: "the diffusion-based audio upsampling"**
>
> The phrase "diffusion-based audio upsampling" refer specifically to the **AudioSR** baseline, which we use as a baseline throughout the paper illustrated in the middle part of Figure 2. We will add more explanations in the final version.
>
> ---
>
> #### **Q6: L145: Clarify "which has been aligned with"**
>
> Thank you for pointing this out. Here, "which has been aligned with" means that our boundary distributions, namely prior $z_ T^{LR}$ and target $z_ 0^{HR}$, have been aligned with the *LR-to-HR* task. We will make our expression more clarified in the final version.
>
> ---
>
> #### **Q7: L155: "while it has already provided", I don't understand this sentence. What does "it" refer to here?**
>
> Thank you for pointing this question out. As shown in our response to **Q1**, the term "it" here refers to the **effective cut-off frequency**, as defined in $ \text{SR}_ {x_ {\text{HR}}} $ in Section 3.2 and denoted as $ f_ {\text{eff}} $ with explanation in Appendix A.1. We will make our expression here clearer in the final version.
>
> ---
>
> #### **Q8: Length of HR and LR signals and potential upsampling artifacts**
>
> Thank you for your valuable suggestion. The LR signals before inference are sampled at a lower rate and therefore have shorter lengths. Figure 2 illustrates the training process of our method, in which we simulate the upsampled low-resolution waveform by directly applying a low-pass filter to the high-resolution signal. In contrast, during inference, to ensure consistency in input-output alignment and support effective inference, we first upsample the LR waveform to the target resolution by **linear upsampling** before feeding it into the model as outlined in line 231.
>
> As you noted, directly performing this step may introduce artifacts, however, our method **addresses** this with a **degradation-aware condition signal $ f_ {\text{prior}} $** (see Section 3.2), which allows the model to **adaptively identify the effective frequency range of the input signal** and thereby mitigate the influence of such artifacts. We visualize this behavior in **Appendix A.2, Figure 4**, which shows how $ f_ {\text{cond}} $ effectively help the model avoid artifacts during inference.
>
> ---
>
> #### **Q9: L164: Define the acronym DiT**
>
> Thank you for pointing this out. "DiT" stands for **Diffusion Transformer** [6] which serves as our score estimator. We will add it in the final version.
>
> ---
>
> #### **Q10: L166: why plural: "bridge models"? are several models trained?**
>
> The use of the plural form "bridge models" is intentional. In our framework, we adopt the latent scaling factor in each stage of our **cascaded bridge models** in Section 3.3, each responsible for performing super-resolution between three successive sampling rates. We use the plural form to indicate the cascading process.
>
> ---
>
> #### **Q11: Inference computational cost**
>
> Thank you for your valuable suggestion. Following your suggestion, we report the real-time factor (RTF) on an NVIDIA-A800 and several baselines under the 48kHz setting as follows:
>
> | Method                  | Modeling Space | Method Type              | RTF ↓  | NFE |
> |-------------------------|----------------|---------------------------|--------|-----|
> | **Ours (any → 48 kHz)**   | Wav-VAE        | Bridge                   | 0.369 | 50  |
> | **Ours (any → 96 kHz)**  | Wav-VAE        | Cascaded Bridges         | 0.695 | 100  |
> | **Ours (any → 192 kHz)**  | Wav-VAE        | Cascaded Bridges         | 1.351 | 150  |
> | Bridge-SR [7] 48 kHz          | Waveform       | Bridge                   | 1.670 | 50  |
> | UDM+ [8]  48 kHz              | Waveform       | Unconditional Diffusion  | 2.320 | 100 |
> | AudioSR [9]     48 kHz        | Mel-VAE        | Conditional Diffusion                | 0.948 | 50 |
> | Fre-painter [10]   48 kHz      | Mel            | GAN                      | 0.009 | 1 |
> | NVSR [11]  48 kHz              | Mel            | GAN                      | 0.033 | 1 |
>
> It is worth noting that under the 48 kHz setting, although our method is not as fast as GAN-based approaches such as [10] and [11], because of the iterative sampling nature, it has significantly outperformed other iterative-based methods including [7], [8], and previous SoTA system AudioSR[9].
>
> When scaling to higher sampling rates (e.g., 192 kHz), it naturally leads to slower inference speeds. Especially, we mitigate the impact by adopting a lighter network architecture, as shown in *Appendix G.2*, so the inference speed does not increase linearly. As shown, even upsampling to 192 kHz, it is faster than the 48 kHz waveform-domain bridge system [7].
>
>
> ---
> [1] Martin Link. Digital audio at 96 khz sampling frequency-pros and cons of a new audio technique. Audio Engineering Society, 1999
>
> [2] Richard Black. Anti-alias filters: the invisible distortion mechanism in digital audio? Audio Engineering Society, 1999.
>
> [3] Joshua D Reiss. A meta-analysis of high resolution audio perceptual evaluation. Journal of the Audio Engineering Society, 2016
>
> [4] Jens Blauert. Spatial hearing: the psychophysics of human sound localization. MIT press, 1997.
>
> [5] J Robert Stuart. Coding high quality digital audio. Japan Audio Society, 1997.
>
> [6] Scalable diffusion models with transformers. ICCV, 2023
>
> [7] Bridge-SR: Schrödinger bridge for efficient sr. ICASSP, 2025
>
> [8] Conditioning and sampling in variational diffusion models for speech super-resolution. ICASSP, 2023
>
> [9] AudioSR: Versatile audio super-resolution at scale. ICASSP, 2024
>
> [10] Audio super-resolution with robust speech representation learning of masked autoencoder. TASLP, 2024
>
> [11] Neural vocoder is all you need for speech super-resolution. Interspeech, 2022
>
> ---

---

> > ### Comment · Reviewer_sHec · 2025-08-05
> >
> > I thank the authors for their clear and detailed responses. I stand by my recommendation of 5: Accept, which I think is well-calibrated for this submission.

---

> > > ### Author Response · Authors · 2025-08-06
> > >
> > > Thank you for recognizing our work. We greatly appreciate your valuable suggestions, and we will make revisions accordingly in the revised version. Thank you again for your effort in the review and the discussion!

---

### Official Review · Reviewer_A2px · 2025-07-04

**Clarity:** 3
**Significance:** 3
**Originality:** 3
**Rating:** 4
**Confidence:** 3

**Summary:**

This paper introduces AudioLBM, a novel audio super-resolution system based on latent bridge models. The core idea is to perform super-resolution as a latent-to-latent transformation. The work proposes frequency-aware training to enable "any-to-any" upsampling and a cascaded architecture with novel prior augmentation strategies to achieve state-of-the-art results and extend super-resolution to ultra-high sampling rates like 96kHz and 192kHz.

**Questions:**

1. The VAE's quality is fundamental to your method's performance. Could you please provide details on its architecture and its standalone reconstruction performance (e.g., SNR or LSD) to help us better understand the upper bound of your system?
2. In the introduction, *"As shown in Fig. 2, AudioSR [38] synthesizes the missing part of the mel-spectrogram latent representation from an uninformative Gaussian prior, ignoring the fact that the LR waveform contains informative cues about the HR target."* In contrast, Transformer or U-Net models can leverage the information in the LR waveform through self-attention and cross-scale feature aggregation mechanisms.
3. In Section 3.1, *"...rather than suffering from area removal in a latent space compressed by the STFT representation or mel-spectrogram [66]."* Could you provide a more detailed explanation of what "area removal" means in the context of STFT?
4. It is unclear why $z^{HR}$ and $z^{LR}$ have the same length $l$. Intuitively, the high-resolution waveform should yield a longer representation than the low-resolution counterpart for the same temporal duration. Could the authors clarify this?

**Ethical Concerns:**

["NO or VERY MINOR ethics concerns only"]

**Final Justification:**

Based on the authors' feedback in rebuttal, some of my concerns of the paper are addressed. So I raise the score.

**Limitations:**

yes

**Quality:**

3

**Strengths And Weaknesses:**

Strength
1.   This paper propose an effective method for audio super-resolution at ultra-high sampling rates (up to 192 kHz), pushing the boundaries of audio SR and enabling professional applications such as audio mastering and post-production.
2. The method demonstrates state-of-the-art performance across a wide range of audio types, including speech, music, and general audio. 3. The comprehensive experiments on multiple benchmark datasets (VCTK, ESC-50, SDS) show consistent and significant improvements over strong baselines in both objective (LSD, SSIM) and subjective (ViSQOL, SigMOS) metrics.

Weakness
1. The paper omits crucial details about the VAE autoencoder, such as its architecture and reconstruction performance, which is a core component and limits reproducibility.
2. The motivation for using a bridge model is not well-justified against other method such as standard conditional diffusion model.
3. The paper makes questionable claims about prior work, such as stating that AudioSR "ignores" informative cues or that spectrogram-based methods suffer from "area removal," without sufficient explanation.
4. Figure 2 does not clearly indicate which method corresponds to the proposed approach and which one represents AudioSR.

---

> ### Author Rebuttal · Authors · 2025-07-30
>
> We thank the reviewer for the valuable comments. We provide a point-to-point response as follows.
>
> ### **Main Comments:**
> ---
> #### **W1 and Q1: The VAE's quality**
>
> We agree that the training and performance of the VAE are central to our method. Due to the limited space of the main paper, we have provided details on its architecture and its standalone reconstruction performance in **Appendix**.
>
> Specifically, we examine the **reconstruction** performance and its corresponding **generation** performance by training our model for 200K steps **on each set of parameters**:
>
> - In **Appendix B.1**, we outline our VAE architecture and loss function, which closely follow the design and training pipeline of Stable-Audio-Open [1] and ETTA [2].
> - In **Appendix B.1.1**, we conduct a systematic exploration of compression rates tailored to the super-resolution task, ultimately choosing a 8× compression ratio, mapping the waveform $ \mathbf{x} \in \mathbb{R}^L $ to $ \mathbf{z} \in \mathbb{R}^{L/512 \times 64} $.
> - In **Appendix B.1.2**, we further investigate the influence of the KL divergence weight, providing both quantitative results and qualitative insights.
> - In **Appendix B.1.3**, we include a detailed comparison of our VAE-based reconstructions with recent, open-sourced reconstruction baselines on 48kHz signals, as shown in **our four tables in the Appendix B**:
>
>
> | Frame Rate     | Model                | SSIM ↑ | LSD ↓ | LSD-LF ↓ | LSD-HF ↓ |
> |--------|----------------------|--------|--------|-----------|-----------|
> |        | **VCTK (48 kHz)**    |        |        |           |           |
> | 100 Hz | agc[3]  (continuous) | 0.913  | 0.762  | 0.766     | 0.741     |
> | 50 Hz  | agc[3]  (discrete)   | 0.917  | 0.789  | 0.754     | 0.768     |
> | 150 Hz | encodec[4]           | 0.887  | 1.126  | 0.761     | 1.176     |
> | 75 Hz  | audiodec[5]          | 0.922  | 0.939  | 0.944     | 0.937     |
> | 75 Hz  | flowdec[6]           | 0.925  | 0.619  | 0.560     | 0.630     |
> | 100 Hz | **Ours**             | **0.951** | **0.580** | **0.428** | **0.602** |
>
> | Frame Rate     | Model                | SSIM ↑ | LSD ↓ | LSD-LF ↓ | LSD-HF ↓ |
> |--------|----------------------|--------|--------|-----------|-----------|
> |        | **ESC-50 (44.1 kHz)**|        |        |           |           |
> | 100 Hz | agc  (continuous) | 0.730  | 0.836  | 0.748     | 0.840     |
> | 50 Hz  | agc  (discrete)   | 0.741  | 0.852  | 0.703     | 0.862     |
> | 150 Hz | encodec          | 0.704  | 0.925  | 0.910     | 0.922     |
> | 100 Hz | dac              | 0.734  | 0.799  | 0.735     | 0.801     |
> | 75 Hz  | audiodec         | 0.695  | 0.999  | 0.969     | 0.997     |
> | 75 Hz  | flowdec          | 0.717  | 0.899  | 0.810     | 0.908     |
> | 100 Hz | **Ours**             | **0.789** | **0.763** | **0.544** | **0.785** |
> - **Appendix B.2** covers the VAE training at higher resolutions (96kHz and 192kHz).
>
> ---
>
> #### **W2 and Q2: Generation process and model architecture**
>
> We would respectfully clarify that our discussion concerns the conceptual difference between **"prior"** and **"condition"** in the context of probabilistic generative processes, which respectively correspond to the **boundary distribution of generative process** and the **auxiliary indicative input of network network**.
>
> Exploiting informative **prior distribution** with bridge models allows the sampling process to start from clean representations $\mathbf{z}_ {T,\text{bridge}}$ that are more instructive to the target than the noisy representations $\mathbf{z}_{T,diff} \sim \mathcal{N}(0, \mathbf{I})$ in conditional diffusion models, therefore reducing the burden of generation process and improving the generation results [7,8,9]. The **condition information** is provided as auxiliary model input, which can control both the noise-to-data generation direction of conditional diffusion models and the data-to-data direction of bridge models [10]. Hence, we respectfully clarify that, **prior** and **condition** are two different techniques for generative models, and improve the generation results from different perspectives. More details can be visited in following references: bridge models [7,8,9], and conditional diffusion models [10].
>
> In audio super-resolution (i.e., **LR-to-HR**) task, as the LR observation has already provided instructive information for the HR target, we naturally improve previous **noise-to-data** generation process with the **data-to-data** one in bridge models, aligning our **boundary distributions** with the **LR-to-HR** task. Namely, both AudioSR and our AudioLBM contain condition information ,i.e., the LR obversation. Hence, the advantages of our method lie in **the constructed transition trajectory between LR and HR pairs**, which fully exploits the informative **LR prior**, rather than the external **condition** that encoded by specific model architecture.
>
> Previously, in AudioSR, the prior $ \mathbf{z}_ {T,diff} $ is drawn from a standard Gaussian distribution $\mathcal{N}(0, I)$ which is not informative to the HR target. The diffusion trajectory is defined as:
> $
> \mathbf{z}_ {t,diff} = \alpha(t) \cdot \mathbf{z}_ 0 + \beta(t) \cdot \boldsymbol{\epsilon}, \boldsymbol{\epsilon} \sim \mathcal{N}(0, \mathbf{I}),
> $
> where the sampling process starts from $\mathbf{z}_ {T,diff} \approx \boldsymbol{\epsilon}$ and **can not exploit instructive LR as prior for generative process**, forcing the model to reconstruct the full HR target purely via **sampling from noise**.
>
> In contrast, our AudioLBM uses a bridge process, where the trajectory is explicitly defined between informative latent variables. Specifically, our formulation is:
> $
> \mathbf{z}_ {t,bridge} = \frac{\alpha_ t \bar{\sigma}_ t^2}{\sigma_ 1^2}  \mathbf{z}_ 0+ \frac{\bar{\alpha}_ t  \sigma_ t^2}{\sigma_ 1^2}  \mathbf{z}_ {T,bridge} + \frac{\alpha_ t  \bar{\sigma}_ t  \sigma_ t}{\sigma_ 1}  \boldsymbol{\epsilon}, \boldsymbol{\epsilon} \sim \mathcal{N}(0, \mathbf{I})
> $.
> Here, the intermediate latent representations $ \mathbf{z}_ {t,bridge} $ has been a stochatisc interpolation between the HR latent $ \mathbf{z}_ 0 $ and the LR latent $ \mathbf{z}_ {T,bridge} $, which allows the generative process to **exploit the informative LR latent**.
>
> Therefore, despite the ability of architectures like Transformer or U-Net to extract and utilize low-frequency **condition**, their **generative process uses uninformative Gaussian prior**, which may lead to sub-optimal performance on audio super-resolution task. Namely, their sampling initiates denoising from Gaussian noise, resulting in a **noise-to-data** generation process whose **boundary distributions** are misaligned with the **LR-to-HR** up-sampling task.
>
> We will clarify this with more explanations in the final version.
>
> ---
>
> #### **W3 and Q3: Explanation of "area removal"**
>
> Thank you for point that out. We would respectfully clarify that the term "area removal" is leveraged from recently proposed STFT-domain bridge model $A^2SB$ [11], refering to the phenomenon where high-frequency components are effectively filtered when using STFT or mel-spectrogram representations.
>
> To elaborate, audio super-resolution aims to recover high-frequency components that are missing from low-resolution waveforms. In the frequency domain (e.g., STFT or mel-spectrogram), this typically manifests as blank or attenuated regions in the upper part of the spectrum, as illustrated in **Line 51** of our submission and reference **[12]**.
>
> We will add more explanations to make this clearer in the final version.
>
>
> ---
>
> #### **W4 and Q4: Explanation of Figure 2, $z^{HR}$ and $z^{LR}$**
>
> Figure 2 illustrates the training pipeline, in which the HR waveform is passed through a low-pass filter to simulate the LR signal. This filtering process preserves the *temporal duration* of the waveform, meaning that both HR and LR waveforms have the **same length** during training.
> To clarify:
> - The **top part** of Figure 2 illustrates how the **low-resolution waveform is simulated during training** using low-pass filtering.
> - The **middle part** corresponds to the **baseline method AudioSR**, which synthesizes high-resolution target from uninformative Gaussian noise.
> - The **bottom part** presents our proposed method, **AudioLBM**, which defines a bridge-based trajectories between informative LR and HR Wav-VAE latents.
>
> Before inference, observed LR representation are typically shorter than the target HR represenatation in length due to lower sampling rate. To bridge this gap between inference and training, we **linearly upsample observed signal to the LR prior in the waveform space**, matching the target HR waveform length before compressing them into the latent space. This ensures temporal alignment and avoids train-test mismatch. This strategy is briefly mentioned in **Section 4.1** ("Training Setup"), and we will add more explanations to clarify this in the final version and make it more explicit in the experiment section to aid understanding of both the training and inference procedures.
>
> ---
> [1] Stable Audio Open. ICASSP, 2025.
>
> [2] ETTA: Elucidating the Design Space of Text-to-Audio Models. ICML, 2025
>
> [3] AudiogenAI. AGC: Audio Generative Compression. Github Repo, 2024
>
> [4] High Fidelity Neural Audio Compression. TMLR, 2023.
>
> [5] Audiodec: An Open-source Streaming High-fidelity Neural Audio Codec. ICASSP, 2023
>
> [6] FlowDec: A Flow-based Full-band General Audio Codec with High Perceptual Quality. ICLR, 2025
>
> [7] Denoising Diffusion Bridge Models. ICLR, 2024
>
> [8] I2SB: image-to-image Schrödinger bridge. ICML, 2023
>
> [9] Schrödinger bridges beat diffusion models on text-to-speech synthesis. arXiv:2312.03491 (2023)
>
> [10] Why are conditional generative models better than unconditional ones?. NeurIPSW, 2022
>
> [11] A2SB: Audio-to-Audio Schrodinger Bridges. arXiv:2501.11311 (2025)
>
> [12] Time-frequency Networks for Audio Super-resolution. ICASSP, 2018
>
> ---

---

> > ### Comment · Reviewer_A2px · 2025-08-02
> > **Feedback from A2px**
> >
> > First of all, I would like to thank the authors for their detailed responses to my previous questions. Some of my concerns have indeed been addressed, but a few issues still remain. I also recognize that my opinions differ from those of other reviewers, so I would appreciate it if the authors could clarify the following points further:
> >
> > ### Regarding Q1 and W1:
> > I apologize for missing the relevant content in the appendix (provided as a separate supplementary file). The use of the VAE encoder-decoder is now clear to me, and the reported performance appears reasonable. I will accordingly raise my score.
> >
> > ### Regarding W2 and Q2:
> > I understand that the use of bridge models may improve generation quality during inference. However, my main concern lies with the statement *"ignoring the fact that the LR waveform contains informative cues about the HR target"*. I believe this is an inaccurate characterization. Prior works—whether via conditioning or through modeling the LR prior—do **not** completely ignore information from the LR signal. This comment feels somewhat overstated, and I hope the authors can rephrase or clarify their intent.
> >
> > ### Regarding W3 and Q3:
> > Could the authors please clarify where in the paper (e.g., which section) it is stated that STFT causes “area removal”? To the best of my knowledge, Mel spectrograms indeed introduce information loss due to frequency compression, but STFT is generally considered a lossless transformation for discrete-time signals (aside from boundary effects). I originally thought this may have been a misstatement in the first draft, but it now seems not. I would appreciate further explanation here, as this directly impacts the correctness of the paper.

---

> ### Author Response · Authors · 2025-08-03
>
> We sincerely thank the reviewer for the thoughtful follow-up and for acknowledging our clarifications regarding the VAE details (W1/Q1), as well as the willingness to raise the score. We greatly appreciate the opportunity to further clarify the remaining points.
>
> #### **W2 and Q2: Motivation of bridge process**
>
> We appreciate your thoughtful feedback and agree that the original wording may have overstated the issue. Our intention was not to imply that prior work entirely ignores information from the LR representation, but rather to highlight a distinction in how this information is utilize for generative trajectory. Specifically, existing approaches often leverage the LR signal for conditioning, but do not explicitly incorporate it into the latent prior distribution itself. Our proposed bridge framework aims to better utilize this information by aligning the LR-informed latent prior with the HR target distribution, thereby enabling more effective learning and sampling trajectories.
> To clarify our point, we will revise the statement in the final version to:
> “While prior methods condition on the LR signal or learn priors independently, they often overlook that the LR waveform carries informative cues that can be integrated into the latent prior. This may lead to sub-optimal probability trajectories during generation.”
>
> #### **W3 and Q3: Explanation of "area removal**
> Thank you for the question. You are right that the STFT is a nearly lossless transform for discrete-time signals. However, our use of the term "area removal" was not meant to suggest information loss caused by the representation (like STFT or Mel) itself, but rather to describe the representational consequence of low-pass filtering in spectral domains, which serves the input signal in our SR task.
> Specifically, in A2SB[1] (Section 3.2, line 5), "area removal" refers to the phenomenon where high-frequency filtering of the waveform manifests as an energy void in the upper region of its spectrogram—visually resembling a blanked-out area. Crucially, this effect stems from the filtering operation, not the spectral transform.
> The same logic applies to mel-spectrograms: the "removed area" reflects the filtering-induced absence of high-frequency energy, not compression from mel-filterbanks. We will clarify this distinction in the final version.
>
> [1] A2SB: Audio-to-Audio Schrodinger Bridges. arXiv:2501.11311 (2025)

---

> ### Comment · Reviewer_A2px · 2025-08-03
> **Feedback from A2px**
>
> I would like to thank the authors for the responses, I have adjusted my score.

---

> > ### Author Response · Authors · 2025-08-06
> >
> > Thank you for your recognition and valuable suggestions. We sincerely appreciate the time and effort you’ve dedicated to providing this feedback.

---

### Decision · Program_Chairs · 2025-09-17

**Decision:**

Accept (poster)

**Comment:**

This paper proposes a method for audio bandwidth extension (“super-resolution”) that supports arbitrary input–output sampling rates. The approach employs conditioned Bridge diffusion in a VAE-derived latent space, and further introduces model cascading to enable reconstruction at ultrahigh frequencies (96 kHz and 192 kHz).

Reviewers found the approach both novel and effective, with performance exceeding state-of-the-art baselines. During the rebuttal, the authors provided additional results and clarifications that successfully addressed all reviewer concerns. I highly encourage the authors to in corporate these into the final manuscript, it would further strengthen this work.

For the above reasons, I recommend acceptance of this submission.